# Population and Co-Occurrence Characteristics of Diagnoses and Comorbidities in Coronary Artery Disease Patients: A Case Study from a Hospital in Guangxi, China

**DOI:** 10.3390/bioengineering11121284

**Published:** 2024-12-18

**Authors:** Jiaojiao Wang, Zhixuan Qi, Xiliang Liu, Xin Li, Zhidong Cao, Daniel Dajun Zeng, Hong Wang

**Affiliations:** 1State Key Laboratory of Multimodal Artificial Intelligence Systems, Institute of Automation, Chinese Academy of Sciences, Beijing 100190, China; zhidong.cao@ia.ac.cn (Z.C.); dajun.zeng@ia.ac.cn (D.D.Z.); 2Cornell Tech, Cornell University, New York, NY 10044, USA; zq83@cornell.edu; 3Faculty of Information Technology, Beijing University of Technology, Beijing 100124, China; liuxl@bjut.edu.cn; 4School of Computer Science, Beijing Institute of Technology, Beijing 100081, China; xinli@bit.edu.cn; 5Department of Cardiology, The People’s Hospital of Guangxi Zhuang Autonomous Region, Nanning 530021, China

**Keywords:** coronary artery disease, diagnoses, comorbidities, network analysis, sensitivity analysis, co-occurrence patterns

## Abstract

Coronary artery disease (CAD) remains a major global health concern, significantly contributing to morbidity and mortality. This study aimed to investigate the co-occurrence patterns of diagnoses and comorbidities in CAD patients using a network-based approach. A retrospective analysis was conducted on 195 hospitalized CAD patients from a single hospital in Guangxi, China, with data collected on age, sex, and comorbidities. Network analysis, supported by sensitivity analysis, revealed key diagnostic clusters and comorbidity hubs, with hypertension emerging as the central node in the co-occurrence network. Unstable angina and myocardial infarction were identified as central diagnoses, frequently co-occurring with metabolic conditions such as diabetes. The results also highlighted significant age- and sex-specific differences in CAD diagnoses and comorbidities. Sensitivity analysis confirmed the robustness of the network structure and identified clusters, despite the limitations of sample size and data source. Modularity analysis uncovered distinct clusters, illustrating the complex interplay between cardiovascular and metabolic disorders. These findings provide valuable insights into the relationships between CAD and its comorbidities, emphasizing the importance of integrated, personalized management strategies. Future studies with larger, multi-center datasets and longitudinal designs are needed to validate these results and explore the temporal dynamics of CAD progression.

## 1. Introduction

Coronary artery disease (CAD) continues to impose a significant global health burden, contributing substantially to morbidity and mortality [1]. The study of CAD and its associated comorbidities is critical for improving patient outcomes, as these comorbidities often complicate disease management. A comprehensive understanding of the multidimensional epidemiological characteristics of CAD patients has become increasingly important in healthcare systems that are shifting toward personalized and integrated care models [2,3]. Epidemiological studies have established that CAD is influenced by a variety of factors, including age, sex, lifestyle, and genetic predisposition, which determine not only disease onset but also its progression and outcomes [4,5,6,7,8]. Gender and age differences in CAD presentation and outcomes are well-documented in the literature [9,10,11,12,13,14,15] and were a key motivation for this analysis. Understanding these differences is critical for developing tailored management strategies.

Another major challenge in managing CAD is the high prevalence of comorbidities, such as hypertension, diabetes, and dyslipidemia, which exacerbate disease severity and increase the risk of adverse cardiovascular events [16,17]. These comorbidities often co-occur in complex patterns, complicating both diagnosis and treatment. Multidimensional epidemiological analysis offers deeper insights into how these factors interact in CAD patients, enabling more accurate risk stratification and personalized care [18]. For instance, research has consistently shown that men are more likely to develop CAD at a younger age and experience myocardial infarctions, while women often present with stable angina later in life [19]. Age-specific trends reveal that older adults are more likely to have multiple comorbidities, which complicates not only the management of CAD but also the associated systemic conditions [20,21].

Complex network analysis has emerged as a powerful tool for studying the relationships between diseases, particularly in the context of multimorbidity [22,23,24,25]. Network-based approaches allow researchers to visualize and quantify the co-occurrence of diseases, providing a holistic view of the interactions between CAD and its comorbidities [26]. Studies have shown that hypertension, diabetes, and metabolic disorders form central hubs in the disease co-occurrence network, reflecting their strong associations with CAD [27]. These findings highlight the importance of addressing not only the primary cardiovascular condition but also the broader network of related comorbidities. Network modularity analysis has been particularly useful in identifying distinct clusters of comorbidities that tend to co-occur with specific CAD diagnoses, a pattern that has important implications for clinical management [28].

Despite significant advances in network medicine, several challenges persist. First, many studies are limited by the lack of longitudinal data and multi-center studies, which impedes the understanding of disease progression and the dynamic nature of comorbidities [29]. Most network-based studies focus on static co-occurrence patterns, overlooking the temporal relationships between diagnoses and comorbidities, which are crucial for understanding the evolution of disease [30,31]. Additionally, geographic variability in CAD risk and comorbidity patterns complicates the generalizability of findings. For example, regional disparities in cardiovascular disease mortality and risk factors have been documented in both high- and low-income countries, underscoring the need for more localized studies [32,33].

To address these gaps, this study focuses on CAD patients from a hospital in Guangxi, China, analyzing the co-occurrence of diagnoses and comorbidities using complex network analysis. The aim is to provide a localized perspective on the multidimensional epidemiological characteristics of CAD patients, identifying key diagnostic clusters and comorbidity hubs that may inform more targeted and effective treatment strategies. This study contributes to the literature by offering new insights into the structure and modularity of comorbidity networks in a specific population, which can serve as a model for other regional analyses. Moreover, by using network-based methods, this research aids in the understanding of disease clustering dynamics and their implications for personalized care.

## 2. Materials and Methods

### 2.1. Experimental Design

This retrospective study was designed to analyze clinical data from patients diagnosed with coronary artery disease (CAD) to identify prevalent diagnoses and comorbidities, assess their associations, and explore underlying network relationships. The study encompassed data collection, preprocessing, statistical analysis, and network analysis to achieve these objectives. A comprehensive flowchart outlining the experimental design is provided in Figure 1, illustrating the key components: data collection, preprocessing, statistical analysis, and network analysis.

### 2.2. Study Population

The baseline dataset comprised 195 patients diagnosed with CAD who were admitted between 2013 and 2020 to the Department of Cardiology at the People’s Hospital of Guangxi Zhuang Autonomous Region. These records originated from 13 cities in Guangxi, with a majority of 66% (*n* = 129) of the patients from Nanning. The 195 CAD patients with full follow-up information have been diagnosed according to authoritative professional guide [34,35,36], and CAD patients without full follow-up information were not included in this dataset. In this study, each medical record contained 7 feature attributes, including hospitalization ID, diagnoses, comorbidities, sex, age, admission year, and city of residence. This comprehensive dataset is suitable for applications in clinical research, patient management, and medical quality assessment. Patient privacy was strictly maintained by anonymizing all identity-related information prior to analysis, adhering to stringent privacy preservation standards.

### 2.3. Data Normalization

The clinical diagnoses in the original electronic medical records (EMRs) lacked standardized terminology, which posed challenges for accurate and consistent analysis. To address this, a natural language processing (NLP) technique was employed alongside custom-developed Python scripts for Chinese text processing and mining. The data normalization and standardization process has been confirmed and verified with professional clinical doctors, which involved several steps to ensure consistency and accuracy. First, manual resolution of textual ambiguities and synonyms was performed; for example, diagnostic entries such as “triple vessel disease”, “left main disease”, “triple coronary disease”, and “multivessel disease” were consolidated under the standardized term “coronary multivessel disease”. Similarly, various descriptions of hypertension, including “hypertension stage 3 very high-risk group”, “hypertension stage 2 high-risk group”, and others, were unified under the general term “hypertension”. Additionally, typographical errors and inconsistencies in the original records were corrected, and duplicate entries were removed through deduplication preprocessing. These steps aimed to ensure the accuracy and consistency of diagnosis and comorbidity classifications while preserving the integrity of the original patient records. Ultimately, all medical diagnostic and comorbidity information was converted to standardized terms, facilitating streamlined manipulation and analysis. The diagnosis and comorbidity information for each patient was complete. From the original text records of diagnoses and comorbidities, we initially extracted 57 distinct diagnosis descriptions and 272 comorbidity descriptions. Due to occasional typographical errors in the original records, we merged similar descriptions of diagnoses and comorbidities based on clinicians’ expert opinions. After this preprocessing step, we obtained 32 unique diagnosis descriptions and 52 comorbidity-related descriptions.

### 2.4. Statistical Analysis

The prevalence of diagnoses and comorbidities among CAD patients was quantified by counting their occurrences in electronic medical records. The detection rate (DR) for each diagnosis and comorbidity was calculated as the ratio of the number of specific diagnoses or comorbidities to the total number of CAD-related records, following the approach outlined in Reference [37], expressed as a percentage:DR_dia/com_ = (N_dia/com_/N_CAD_) × 100%(1)

Sex-specific detection rates were determined by calculating the ratio of each diagnosis or comorbidity within male and female subgroups relative to the total number of CAD patients in each sex group. Odds ratios (OR) and their 95% confidence intervals (CI) were computed for each sex-specific DR to assess the strength of associations between diagnoses/comorbidities and sex. For age-specific analyses, patients were categorized into 10-year age intervals (e.g., 35–44 years, 45–54 years) ranging from 35 to 88 years. Within each age group, DRs were calculated along with corresponding 95% CIs. Data preprocessing and normalization were performed using Python scripts for text processing, mining, and statistical analysis. These comprehensive statistical methods ensure the reproducibility and verification of results by knowledgeable readers with access to the original data.

### 2.5. Network Analysis

To explore the co-occurrence relationships between diagnoses and comorbidities among CAD patients, we used the preprocessed diagnosis and comorbidity descriptions to construct two types of weighted co-occurrence networks [38,39], incorporating both homogeneous and heterogeneous nodes [40]. The first type is a monopartite graph consisting of nodes of the same type (diagnosis or comorbidity) and the second type is a bipartite graph consisting of nodes of different types (diagnosis and comorbidity). A co-occurrence relationship was established when two diagnoses or comorbidities appeared together in a single electronic medical record. The frequency of these co-occurrences served as the weight of the relationship. In this network, nodes represented individual diagnoses or comorbidities, with node sizes proportional to their frequencies within the dataset. Edges between nodes indicated co-occurrence relationships, with edge weights corresponding to the number of times the paired diagnoses or comorbidities co-occurred across all records. In cases where a single medical record included multiple comorbidities, each possible pair within that record received an increment in their relationship count. While the sample size in this study is relatively small, we implemented several strategies to ensure the robustness of the network analysis, including the use of weighted networks to enhance reliability, and cross-validation with existing literature. These steps mitigate the limitations associated with small datasets and support the validity of the findings.

Several network metrics were employed to assess the importance of nodes within the network [41]. Degree centrality [42], the total number of direct connections a node has, was calculated to identify diagnoses or comorbidities with higher interconnectedness. The average degree of the network provided an overall measure of connectivity, while the average path length measured the average number of steps along the shortest paths for all possible pairs of nodes, indicating the closeness of relationships within the network. The four metrics—average clustering coefficient [43], modularity [44], description length [45,46], and betweenness centrality [42,47]—were used to offer complementary insights into the structural and functional properties of the co-occurrence networks. The clustering coefficient emphasizes local cohesiveness, modularity evaluates global community structure, and betweenness centrality identifies key nodes for connectivity and flow. Together, they provide a comprehensive framework for network analysis. A force-directed layout algorithm [48] was used to visualize the network, positioning nodes based on their connections to enhance interpretability. Diagnoses or comorbidities with stronger and more frequent relationships tended to cluster together, facilitating the identification of significant patterns and associations within the CAD patient population. Network analysis and visualizations were conducted using specialized software such as Gephi 0.10.1 [49] and R 4.4.1 [50], ensuring accurate and interpretable results. To strengthen the robustness of the network metrics, sensitivity analyses based on adjusting edge weight filtering thresholds were conducted, validating the reliability of the network structure and modularity findings. Through these network analysis methods, the study aimed to identify significant patterns and associations within the CAD patient population, providing insights into the interconnectedness of various health conditions.

## 3. Results

### 3.1. Age and Sex Distribution of CAD Patients

Figure 2 illustrates the age distribution of hospitalized coronary artery disease (CAD) patients stratified by sex. The density curve for males peaks around the age of 60, indicating a higher concentration of male patients in their 50s and 60s. In contrast, the female distribution curve exhibits a broader and slightly later peak, with the majority of female patients concentrated between the ages of 60 and 70. This suggests that males are hospitalized for CAD at younger ages compared to females, who are more frequently affected at older ages. Despite differing sample sizes (138 males and 57 females), the density plot standardizes the data using probability density, enabling better visualization and comparison. Figure 3 further delineates the sex- and age-specific distribution characteristics of the top 20 CAD diagnoses and comorbidities. Unstable angina is prevalent across both sexes, with a higher density in males aged 50–60 and females aged 60–70. Myocardial infarction is predominantly found in males, peaking around 50–60 years, whereas coronary atherosclerosis and coronary atherosclerotic heart disease are more common in females, especially in older age groups. Similarly, Figure 4 depicts the distribution of top comorbidities, highlighting that hypertension is the most prevalent, with higher density in females between 60 and 70 years, while males exhibit a broader peak around 50 to 60 years. These figures collectively highlight significant sex- and age-related differences in the distribution of CAD diagnoses and comorbidities, indicating that males generally exhibit earlier onset of diagnoses, whereas females tend to show later onset with peaks in older age groups.

### 3.2. Detection Rates of Top Diagnoses and Comorbidities

Table 1 lists the top 20 CAD diagnoses with the highest detection rates. Unstable angina had the highest detection rate at 42.05%, followed by Braunwald Class II B with rates exceeding 20%. Diagnoses such as stable angina, myocardial infarction, CCS Angina Class II, and KILLIP Class I each had detection rates above 10%, indicating a greater risk associated with these conditions in CAD patients. Sex-specific detection rates and odds ratios (OR) for these top diagnoses are also presented in Table 1. Myocardial infarction showed the largest difference between males and females, with males having a 94% higher risk (OR: 1.94, 95% CI: 0.73–5.20) compared to females. Conversely, coronary atherosclerosis was slightly more prevalent in females (OR: 0.49, 95% CI: 0.18–1.36). Certain diagnoses, such as KILLIP Class I and CCS Angina Class I, were observed exclusively in males, resulting in infinite odds ratios. However, most confidence intervals included 1, indicating that these sex differences were not statistically significant, likely due to the relatively small sample size.

Table 2 outlines the detection rates and sex distribution of the top 20 comorbidities in CAD patients. Hypertension was the most prevalent comorbidity, affecting 61.54% of patients, with a similar distribution between males and females (OR: 0.66, 95% CI: 0.35–1.27). Metabolic diseases and dyslipidemia had detection rates exceeding 20%, while diabetes and its complications, fatty liver disease, cardiac arrhythmias, renal cysts, and post-percutaneous coronary intervention (PCI) each had detection rates above 10%. Metabolic diseases and fatty liver disease were more common in males, whereas dyslipidemia was notably higher in females (OR: 0.43, 95% CI: 0.21–0.88). Renal cysts were significantly more prevalent in males (OR: 4.06, 95% CI: 1.06–15.64). Similarly to diagnoses, several comorbidities, such as myocardial infarction and tobacco dependence, were observed only in males, resulting in infinite odds ratios. Most odds ratios did not reach statistical significance, as confidence intervals frequently included 1, suggesting that observed sex differences were not conclusive.

### 3.3. Top Five Diagnoses and Comorbidities Across Demographics

Table 3 and Table 4 provide detailed insights into the distribution of diagnoses and comorbidities among CAD patients. These tables reveal that unstable angina consistently emerges as the most prevalent diagnosis and hypertension as the leading comorbidity across most demographic categories. Additionally, the tables highlight notable variations in other conditions based on age, sex, and temporal factors, underscoring the importance of personalized management strategies in CAD care.

#### 3.3.1. Top Five Diagnoses of CAD Across Age Groups, Sexes, and Admission Years

Table 3 elucidates the distribution of the top five CAD diagnoses across different age groups (30–89 years), sexes, and admission years from 2013 to 2020. Across all age categories, Unstable Angina (UA) consistently emerged as the most prevalent diagnosis, maintaining its prominence in both younger (30–39 years) and older (80–89 years) cohorts. Braunwald Class II B frequently followed as the second most common diagnosis, except in the 30–39 and 70–79 age groups, where Braunwald Class III B and Coronary Atherosclerosis (CA), respectively, were more prevalent. Myocardial infarction (MI) was notably prominent in the 40–49 and 50–59 age groups, reflecting a higher incidence among middle-aged males. Conversely, CA showed an increasing trend in older age groups (50–59, 70–79, and 80–89 years), suggesting heightened vulnerability with advancing age. CCS Angina Class II maintained its rank within the top five diagnoses across all age groups, emphasizing its ongoing relevance in CAD assessment and management.

Gender-specific analysis revealed that while UA was the leading diagnosis in both males and females, MI was significantly more frequent in males, positioning it as the third most common diagnosis in this group. In contrast, CA was more prevalent in females, ranking fifth, which may indicate gender-specific pathophysiological differences in CAD manifestation. These patterns are consistent with existing studies that highlight hormonal influences and differing risk factor profiles between sexes [51,52,53,54]. Yearly trends indicated that UA remained the foremost diagnosis throughout the study period. However, MI exhibited peaks in 2013 and 2020, potentially reflecting shifts in diagnostic practices or demographic changes. CCS Angina Class II and CA demonstrated increased frequencies in the latter years (2017–2020), possibly due to enhanced diagnostic techniques or evolving patient populations.

The consistent prevalence of UA underscores its critical role in CAD pathophysiology, serving as a key indicator for impending cardiac events and guiding urgent clinical interventions. The observed age- and gender-specific variations in other diagnoses highlight the necessity for personalized diagnostic and therapeutic strategies in CAD management, ensuring that interventions are tailored to the unique risk profiles and disease presentations of different patient subgroups. The comprehensive analysis of Table 3 underscores the enduring significance of unstable angina across diverse demographics while revealing nuanced trends in other CAD diagnoses based on age, sex, and temporal factors. These insights inform the development of targeted clinical approaches, emphasizing the importance of considering demographic variables in the diagnosis and management of CAD to enhance patient outcomes and optimize healthcare delivery.

#### 3.3.2. Top Five Comorbidities of CAD Across Age Groups, Sexes, and Admission Years

Table 4 presents the top five comorbidities of CAD across different age groups (35–88 years), sexes, and admission years. Hypertension consistently ranked as the most common comorbidity in all age groups except the youngest (30–39 years), where Cerebral Infarction (CI) was the most prevalent. This exception highlights a unique comorbidity profile in younger CAD patients, where cerebrovascular events are more prominent compared to older cohorts. In age groups 40–49 and older (50–89 years), hypertension remained the dominant comorbidity, affecting a substantial proportion of patients (e.g., 61.54% overall). Other frequently observed comorbidities included Cardiac Arrhythmias, Dyslipidemia, Metabolic Diseases, and Diabetes and its complications. The prevalence of these comorbidities varied significantly by age and sex. Specifically, younger patients (30–49 years) were more likely to present with fatty liver disease and post-percutaneous coronary intervention (Post PCI), whereas older age groups (70–89 years) exhibited higher prevalence rates of diabetes, atherosclerosis, and stenotic diseases. This trend suggests an age-related increase in metabolic and atherosclerotic conditions among CAD patients.

Sex-specific analysis revealed that among males, fatty liver disease was more prevalent, whereas dyslipidemia was notably higher in females. This gender disparity indicates potential differences in risk factor profiles and disease progression between sexes. For example, metabolic conditions like fatty liver disease may be more influenced by lifestyle and genetic factors prevalent in males, while dyslipidemia may be more strongly associated with hormonal and metabolic variations in females. Over the study period from 2013 to 2020, hypertension consistently remained the dominant comorbidity across all years. However, there were fluctuations in the prevalence of other comorbidities such as cardiac arrhythmias, diabetes, and dyslipidemia. Similarly, the occurrence of atherosclerosis and stenotic diseases increased in later years, which may be attributed to improved diagnostic capabilities or evolving patient demographics.

These findings underscore the persistent association of hypertension with CAD, reinforcing its central role in the pathophysiology and management of the disease. The observed age- and sex-specific patterns in other comorbidities highlight the necessity for targeted management strategies. For younger patients, emphasis on preventing and managing cerebrovascular events and metabolic conditions may be crucial, while for older patients, addressing diabetes and atherosclerotic complications remains paramount. Additionally, the gender differences in comorbidity prevalence suggest that personalized approaches considering sex-specific risk factors could enhance patient outcomes. The comprehensive analysis of Table 4 highlights hypertension as the most prevalent comorbidity in CAD patients across most demographics, while also revealing distinct age- and sex-specific patterns in other comorbidities. These insights are valuable for informing clinical practices and developing targeted intervention strategies to manage the diverse health profiles of CAD patients effectively.

### 3.4. Co-Occurrence Frequency of Diagnoses and Comorbidities

Table 5 presents diagnosis–comorbidity pairs with a co-occurrence frequency of 10 or more among CAD patients. The pair of unstable angina and hypertension had the highest co-occurrence frequency at 59, indicating a strong association between these two conditions. Other high-frequency pairs include Braunwald Class II B with hypertension (31), stable angina with hypertension (24), unstable angina with dyslipidemia (20), metabolic diseases (17), and fatty liver disease (16). Myocardial infarction frequently co-occurred with hypertension (17), reinforcing the critical role of hypertension in severe cardiovascular events. These findings suggest that comorbid conditions, particularly hypertension and metabolic disorders, are highly prevalent in CAD patients, underscoring the need for comprehensive management of these co-occurring conditions to improve patient outcomes.

Table 6 lists CAD diagnoses that co-occur with 10 or more different comorbidity types. Unstable angina exhibited the highest number of co-occurring comorbidities (45), followed by Braunwald Class II B (40), stable angina (36), myocardial infarction (32), and CCS Angina Class II (31). Other diagnoses with substantial comorbidity co-occurrence include Braunwald Class III B (28), KILLIP Class I (23), and coronary atherosclerosis (22). These results highlight that key CAD diagnoses are frequently accompanied by a diverse range of comorbidities, indicating complex health profiles in these patients that necessitate comprehensive and multifaceted management approaches.

Table 7 presents comorbidities that co-occur with 10 or more different diagnosis types in CAD patients. Hypertension ranks highest, co-occurring with 27 distinct diagnoses, underscoring its central role as a prevalent comorbidity in CAD patients. Metabolic diseases (21), diabetes and its complications (19), and pulmonary infections (19) also show high degrees of co-occurrence with various diagnoses, indicating their frequent involvement in the health profiles of CAD patients. Other significant comorbidities include cardiac arrhythmias (17), renal dysfunction (16), and dyslipidemia (15), all demonstrating strong associations with multiple diagnoses. Conditions such as atherosclerosis and stenotic diseases (14), valvular heart diseases (13), and renal cysts (12) also frequently co-occur with a wide range of diagnoses. These findings illustrate that certain comorbidities, particularly hypertension and metabolic disorders, are highly interconnected with various CAD diagnoses, highlighting the need for integrated management strategies to address the complex health needs of this patient population.

### 3.5. Network Analysis of Diagnoses and Comorbidities

To explore the co-occurrence relationships between diagnoses and comorbidities among CAD patients, comprehensive network analyses were conducted. Figure 5 illustrates the modularity within the co-occurrence network of CAD diagnoses. The modularity class indicates diagnostic clusters or communities where diagnoses are more likely to co-occur. Each color represents a distinct modularity class, highlighting the interplay and grouping of diagnoses that share stronger associations within the network. These clusters provide insights into specific patterns, revealing potential relationships and patterns for targeted clinical management. Unstable angina serves as a central and highly connected node, showing strong associations with Braunwald Class II B, Braunwald Class III B, and myocardial infarction. Other key diagnoses, such as myocardial infarction and coronary atherosclerosis, also exhibit multiple connections, highlighting their co-occurrence with conditions like KILLIP Class I, acute coronary syndrome, and heart failure. Stable angina and CCS Angina Class II form smaller, distinct clusters with weaker connections to other conditions, suggesting specific patterns of disease progression or shared risk factors among CAD patients.

Figure 6 presents the modularity within the co-occurrence network of CAD comorbidities. The modularity class highlights clusters of comorbid conditions that are strongly interrelated or frequently co-occurring in the CAD patient population. Each color in the network visualization corresponds to a distinct modularity class, representing a unique community of comorbidities. This classification serves as a powerful tool for uncovering hidden patterns in the comorbidity network. Hypertension is the most central and highly connected comorbidity, linked to a wide range of other conditions, including cardiac arrhythmias, dyslipidemia, metabolic diseases, and diabetes and its complications. These conditions form a dense network of interrelated comorbidities, indicating frequent co-occurrence in CAD patients. Fatty liver disease, renal dysfunction, and renal cysts also demonstrate strong connections, particularly with metabolic and cardiovascular conditions. Peripheral comorbidities, such as pulmonary infections, skeletal diseases, and urological diseases, are less densely connected but still show notable associations with core comorbidities. The network structure reveals that metabolic and cardiovascular conditions, especially hypertension, serve as central hubs in the co-occurrence of comorbidities in CAD patients, emphasizing the multifaceted nature of disease management in this population.

Figure 7 illustrates a bipartite co-occurrence network connecting diagnoses and comorbidities in CAD patients. In this network, unstable angina and myocardial infarction emerge as central diagnoses, linked to a wide array of comorbidities including hypertension, diabetes and its complications, cardiac arrhythmias, metabolic diseases, and dyslipidemia. These core comorbidities exhibit connections to multiple diagnoses, forming dense clusters of co-occurrence. The network highlights the strong interrelationship between cardiovascular conditions and metabolic comorbidities, such as renal dysfunction, fatty liver disease, and renal cysts, which are frequently observed in CAD patients. Peripheral diagnoses like Braunwald Class III and KILLIP Class I show more specific or limited connections to a smaller set of comorbidities. Overall, the network emphasizes the complex interplay between diagnoses and comorbidities in CAD patients, suggesting that managing both cardiovascular and metabolic conditions is crucial for comprehensive patient care.

### 3.6. Sensitivity Analysis of Co-Occurrence Networks

The study investigated the changes in topological properties and robustness of the diagnosis co-occurrence network, comorbidity co-occurrence network, and diagnosis–comorbidity bipartite network by adjusting edge weight filtering thresholds. The sensitivity analysis of the three networks demonstrates how edge weight filtering affects their topological properties (Appendix A
Table A3). Overall, filtering low-weight edges significantly reduced the number of edges, average degree, and network density while increasing the number of weakly connected components and average path length. These changes indicate that low-weight edges play a critical role in maintaining the connectivity of the networks, but their removal enhances the prominence of stronger connections that better reflect the core structure of the networks. The average clustering coefficient and average weighted degree also decreased with higher filtering thresholds, suggesting that low-weight edges contribute to local clustering and weighted connectivity.

In the diagnosis co-occurrence network (Appendix A Figure A1), edge filtering highlighted the network’s underlying modular structure. While the unfiltered network exhibited a relatively high modularity of 0.61, filtering low-weight edges slightly improved modularity, indicating clearer community structures. The network’s description length decreased with filtering, showing a reduction in complexity and improved interpretability of the resulting communities. Across all filtering thresholds, Unstable Angina consistently emerges as the central node, reinforcing its critical role in CAD diagnosis. Its strong connections to other diagnoses, such as Stable Angina and Myocardial Infarction, highlight its importance in understanding CAD progression and comorbidity patterns. These results suggest that the removal of low-weight edges is effective in identifying core diagnostic relationships, which could better reflect significant co-occurrence patterns among diagnoses rather than noise.

The comorbidity co-occurrence network (Appendix A Figure A2) displayed a similar trend, with modularity increasing from 0.12 in the unfiltered network to 0.22 after higher thresholds were applied. This substantial improvement in modularity highlights the presence of more distinct community structures when low-weight edges, which likely represent weaker or less significant co-occurrence relationships, are removed. The reduction in description length further supports this finding, indicating that the filtered network captures a more concise and meaningful structure. Across all filtering thresholds, Hypertension consistently emerges as the central hub of the network, reflecting its significant role as a primary risk factor and comorbidity in CAD. Its strong associations with conditions such as Dyslipidemia, Diabetes, and Cardiac Arrhythmias highlight the interconnected nature of metabolic and cardiovascular disorders in CAD patients. These results underscore the importance of focusing on high-weight edges to identify key comorbidities associated with coronary heart disease.

In the diagnosis–comorbidity bipartite network (Appendix A Figure A3), edge filtering led to a marked decrease in complexity, as reflected by a significant drop in description length from 1497.60 in the unfiltered network to 500.24 after higher thresholds were applied. Modular structures became more distinct, with modularity increasing from 0.21 to 0.27. Across all filtering thresholds, Unstable Angina and Hypertension consistently appear as central hubs, underscoring their critical roles in CAD diagnosis and management. These nodes are strongly connected to major comorbidities like Diabetes, Dyslipidemia, and Fatty Liver Disease, reflecting the interconnected nature of metabolic and cardiovascular conditions in CAD patients. These findings reveal that low-weight edges in the bipartite network contribute to noise and obscure the primary relationships between diagnoses and comorbidities. By reducing these edges, the network becomes more interpretable, highlighting critical diagnostic–comorbidity associations that could inform targeted clinical interventions and improve our understanding of the interplay between diagnoses and associated conditions.

## 4. Discussion

This study provides a comprehensive analysis of the prevalence and distribution of diagnoses and comorbidities among patients with coronary artery disease (CAD) in Guangxi, highlighting significant age- and sex-specific patterns. The findings reveal that male patients tend to be hospitalized for CAD at younger ages, primarily in their 50s and 60s, whereas female patients exhibit a broader age distribution with a peak between 60 and 70 years. This gender disparity in the age of onset aligns with existing literature, which suggests that hormonal differences and lifestyle factors contribute to the later onset of CAD in females compared to males [55,56].

Unstable angina emerged as the most prevalent diagnosis, affecting over 42% of CAD patients, followed by Braunwald Class II B and stable angina. The high detection rates of these conditions underscore their critical role in the clinical management of CAD. Myocardial infarction was significantly more common in males, with an odds ratio indicating a nearly twofold higher risk compared to females. This gender-specific prevalence is consistent with previous studies that have documented higher incidences of acute myocardial infarction in men, potentially due to factors such as higher rates of tobacco use and more aggressive forms of atherosclerosis [57,58,59,60].

Hypertension was identified as the most prevalent comorbidity, present in over 61% of CAD patients. Its central role within the comorbidity network, linked to various other conditions such as metabolic diseases, dyslipidemia, and diabetes, highlights the multifaceted nature of CAD and the importance of comprehensive management strategies. The co-occurrence of hypertension with multiple comorbidities reinforces the need for integrated care approaches that address both cardiovascular and metabolic risk factors simultaneously [61,62]. The identification of hypertension and metabolic disorders as central hubs highlights opportunities for targeted interventions, such as aggressive blood pressure control and metabolic risk management. These findings also underscore the importance of integrated care models that address the multifaceted nature of CAD comorbidities, paving the way for personalized treatment plans.

The network analysis revealed that hypertension and metabolic disorders act as central hubs within the comorbidity network, suggesting that these conditions are pivotal in the progression and management of CAD. This interconnectedness mirrors findings from previous research, which emphasize the synergistic impact of hypertension and metabolic syndrome on cardiovascular outcomes [63]. While our findings align with previous research, the network analysis approach provides unique insights by identifying central hubs of comorbidities and their interactions. This novel perspective enables a more targeted and integrated approach to managing coronary artery disease, advancing both clinical and epidemiological practices. Furthermore, the bipartite co-occurrence network illustrated complex interactions between diagnoses and comorbidities, reinforcing the necessity for holistic treatment plans that consider the interplay of multiple health conditions in CAD patients.

The edge weight filtering strategy plays a crucial role in refining network structures by removing low-weight edges that may introduce noise or obscure significant connections. This process highlights the core relationships within networks, resulting in increased modularity and reduced network complexity as reflected by lower description lengths. By filtering out weaker connections, co-occurrence networks become more interpretable, with clearer community structures and enhanced modular characteristics. However, this comes at the cost of network sparsity, as filtering reduces edge density, average degree, and clustering coefficients, potentially affecting the overall connectivity and local clustering properties of the network. For practical applications, moderate edge weight filtering is recommended for diagnosis co-occurrence networks to enhance the identification of significant diagnostic patterns while retaining sufficient connectivity. In comorbidity co-occurrence networks, higher thresholds can effectively capture key comorbidities without being influenced by weaker associations. In diagnosis–comorbidity bipartite networks, filtering strategies should balance sparsity and interpretability, focusing on extracting critical diagnostic–comorbidity relationships. These approaches provide valuable insights into the core structures of the networks, enabling more robust and clinically meaningful analyses of disease patterns and associations.

Despite the insightful findings, this study has several limitations that should be acknowledged. First, the dataset was derived from a single hospital in Guangxi, which may not fully represent the broader population. With a sample size of 195 patients who had complete follow-up information, some degree of sampling error is expected. The low number of patients with certain diagnoses, such as heart failure, reflects the strict inclusion criteria, which are directly related to the sample size. Additionally, selection bias is a potential limitation of this study due to the retrospective nature of the dataset, which was collected from a specific population. Patients with milder forms of coronary artery disease or those managed in primary care settings may be under-represented, leading to an over-representation of severe cases and associated comorbidities. These factors may skew the observed co-occurrence patterns and limit the generalizability of the findings to broader or more diverse populations. Second, the retrospective study design may introduce inherent biases arising from incomplete or inconsistent medical records or to have undergone detailed diagnostic evaluations, potentially overemphasizing certain diagnoses or comorbidities. Future studies that incorporate prospective data collection and multi-center datasets would help mitigate these biases, improve generalizability, and provide a more balanced representation of CAD populations. Third, data preprocessing in this study involved a combination of automated NLP techniques and manual standardization. While this approach ensured contextual accuracy, it may have introduced inconsistencies or subjective biases during the manual standardization process. Future studies should aim to incorporate fully automated, validated workflows to enhance reproducibility and accuracy in data processing. Fourth, the cross-sectional nature of this analysis precludes the establishment of causal relationships between diagnoses and comorbidities. The focus on static co-occurrence patterns provides valuable insights into the relationships between CAD and its comorbidities but does not capture the temporal dynamics of disease progression. Incorporating longitudinal data in future studies could reveal the evolution of comorbidities over time and their causal relationships, offering more targeted intervention strategies. Longitudinal studies with larger, more diverse cohorts and prospective designs are essential to validate these findings and explore the underlying mechanisms driving the observed sex- and age-specific patterns in CAD diagnoses and comorbidities. Finally, the findings of this study are based on data from a single region, which may not account for geographical variations in risk factors, healthcare access, or socioeconomic conditions. Future studies should include data from diverse geographic regions to enhance the robustness and generalizability of the conclusions. By addressing these limitations through prospective, multi-center, and longitudinal studies, future research can provide a more comprehensive understanding of the complex interplay between CAD diagnoses and comorbidities, ultimately informing more effective prevention and intervention strategies.

In conclusion, this study enhances our understanding of the complex interplay between various diagnoses and comorbidities in CAD patients, emphasizing the critical roles of hypertension and metabolic disorders. The identified age- and sex-specific patterns provide valuable insights for tailoring clinical management and preventive measures. Our study advances the understanding of coronary artery disease by leveraging network analysis to uncover novel co-occurrence patterns and central hubs, such as the critical roles of Unstable Angina and Hypertension in CAD comorbidity networks. These findings go beyond reaffirming existing literature by providing actionable insights into prioritizing comorbidity management and identifying targets for early intervention strategies, offering a deeper understanding of the complex interplay between diagnoses and comorbidities. Addressing the limitations through future research will further solidify these findings and contribute to improved cardiovascular health outcomes.

## Figures and Tables

**Figure 1 bioengineering-11-01284-f001:**
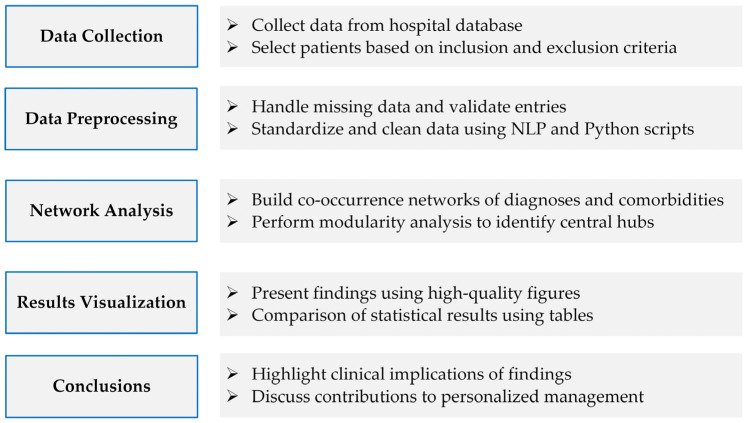
A comprehensive flowchart outlining the experimental design.

**Figure 2 bioengineering-11-01284-f002:**
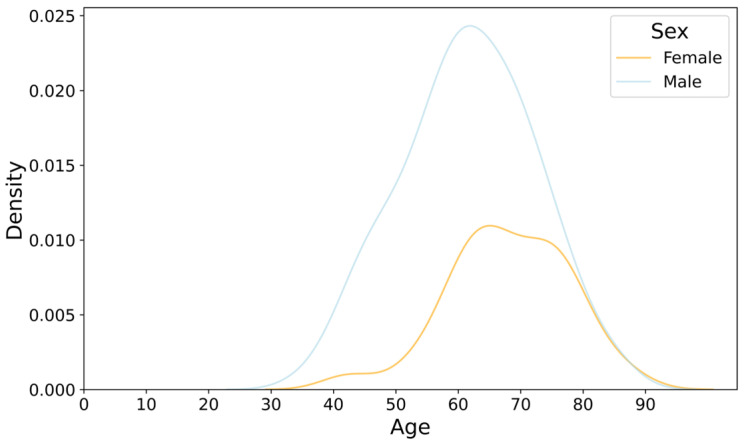
Age Distribution of Hospitalized Coronary Artery Disease (CAD) Patients by Sex in Guangxi.

**Figure 3 bioengineering-11-01284-f003:**
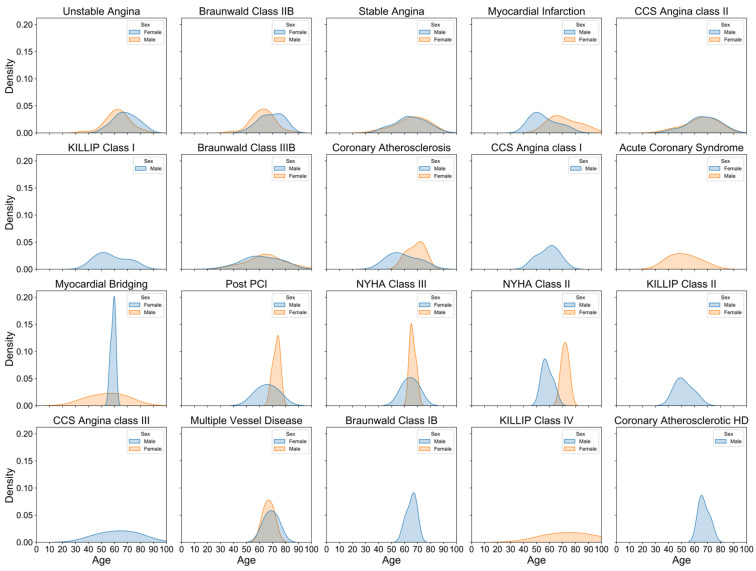
Sex- and Age-Specific Distribution of the Top 20 Diagnoses in CAD Patients.

**Figure 4 bioengineering-11-01284-f004:**
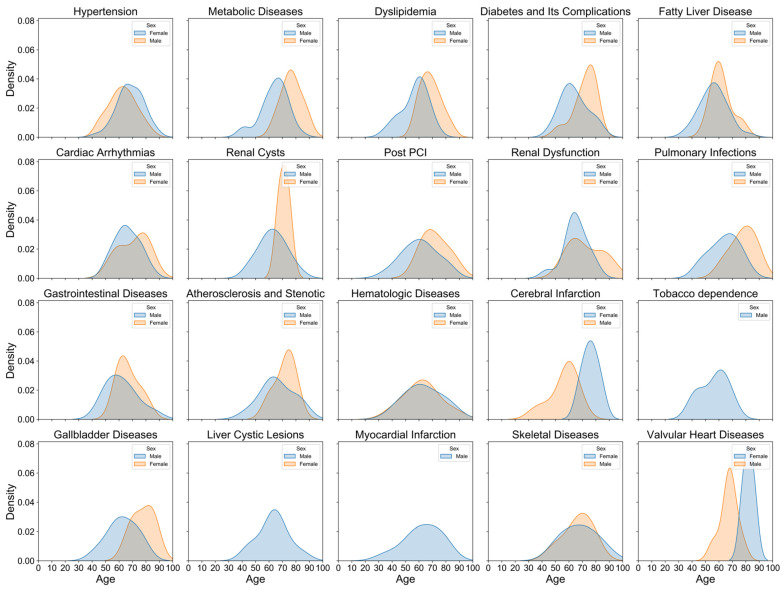
Sex- and Age-Specific Distribution of the Top 20 Comorbidities in CAD Patients.

**Figure 5 bioengineering-11-01284-f005:**
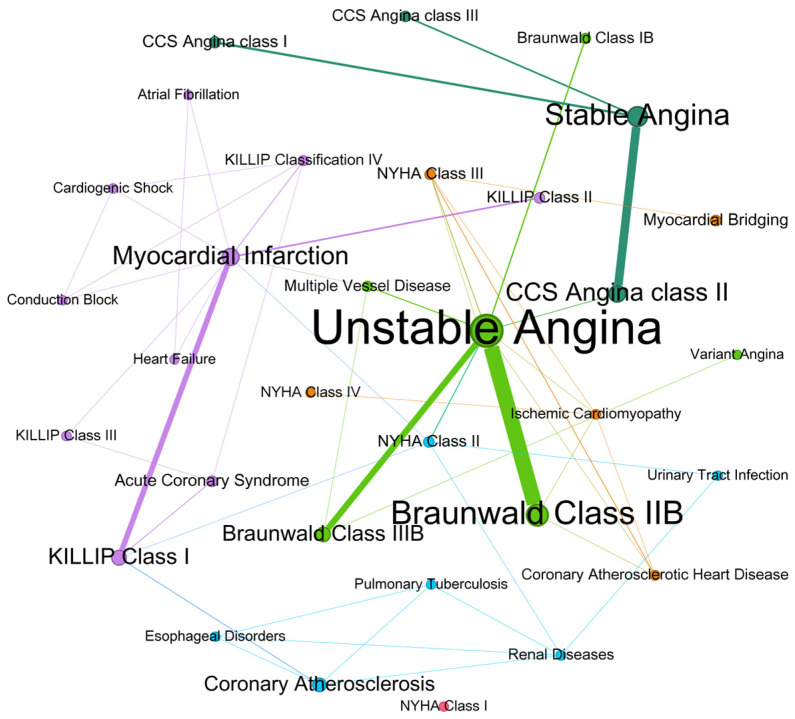
Modularity Classification of the Co-occurrence Network for Diagnoses in CAD Patients.

**Figure 6 bioengineering-11-01284-f006:**
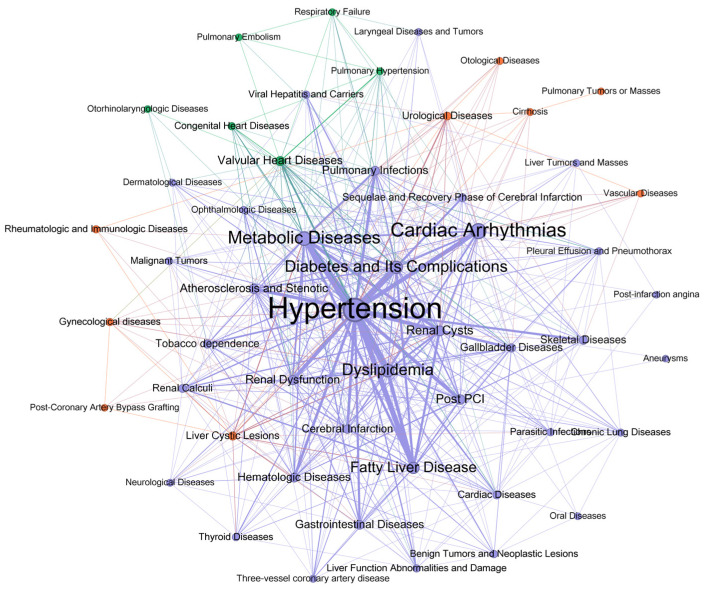
Modularity Classification of the Co-occurrence Network for Comorbidities in CAD Patients.

**Figure 7 bioengineering-11-01284-f007:**
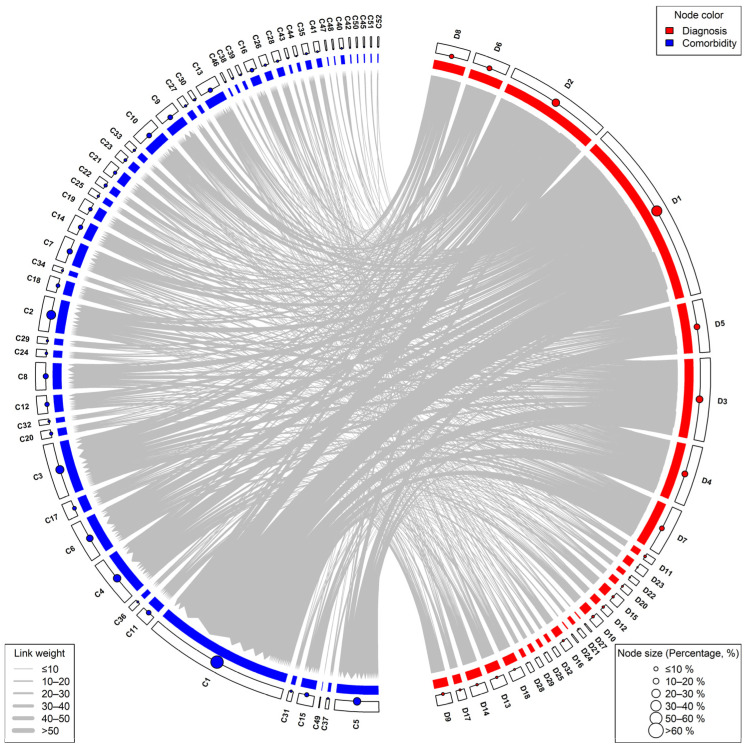
Bipartite Co-occurrence Network of Diagnoses and Comorbidities in CAD Patients. Please refer to Appendix A Table A1 and Table A2 for the names corresponding to the node codes.

**Table 1 bioengineering-11-01284-t001:** Detection Rates and Sex-Specific Distribution of the Top 20 Diagnoses in CAD Patients.

Diagnosis	Count (%)	Male (%)	Female (%)	OR (95% CI)
Unstable Angina	82 (42.05)	55 (39.86)	27 (47.37)	0.74 (0.40, 1.37)
Braunwald Class II B	45 (23.08)	32 (23.19)	13 (22.81)	1.01 (0.49, 2.08)
Stable Angina	38 (19.49)	30 (21.74)	8 (14.04)	1.64 (0.71, 3.76)
Myocardial Infarction	28 (14.36)	23 (16.67)	5 (8.77)	1.94 (0.73, 5.20)
CCS Angina class II	27 (13.85)	20 (14.49)	7 (12.28)	1.16 (0.47, 2.86)
KILLIP Class I	20 (10.26)	20 (14.49)	0 (0.00)	INF *
Braunwald Class III B	19 (9.74)	13 (9.42)	6 (10.53)	0.85 (0.32, 2.29)
Coronary Atherosclerosis	16 (8.21)	9 (6.52)	7 (12.28)	0.49 (0.18, 1.36)
CCS Angina class I	7 (3.59)	7 (5.07)	0 (0.00)	INF *
Post-Percutaneous Coronary Intervention	6 (3.08)	3 (2.17)	3 (5.26)	0.40 (0.09, 1.83)
Acute Coronary Syndrome	6 (3.08)	5 (3.62)	1 (1.75)	1.55 (0.25, 9.69)
Myocardial Bridging	6 (3.08)	3 (2.17)	3 (5.26)	0.40 (0.09, 1.83)
NYHA Class III	5 (2.56)	3 (2.17)	2 (3.51)	0.57 (0.11, 2.99)
NYHA Class II	5 (2.56)	3 (2.17)	2 (3.51)	0.57 (0.11, 2.99)
KILLIP Class II	5 (2.56)	4 (2.90)	1 (1.75)	1.26 (0.19, 8.21)
CCS Angina class III	5 (2.56)	4 (2.90)	1 (1.75)	1.26 (0.19, 8.21)
Multiple Vessel Disease	4 (2.05)	2 (1.45)	2 (3.51)	0.41 (0.07, 2.41)
Braunwald Class I B	4 (2.05)	3 (2.17)	1 (1.75)	0.97 (0.14, 6.76)
KILLIP Classification IV	3 (1.54)	1 (0.72)	2 (3.51)	0.24 (0.03, 1.88)
Coronary Atherosclerotic Heart Disease	3 (1.54)	3 (2.17)	0 (0.00)	INF *

* INF indicates cases where the odds ratio (OR) cannot be calculated due to a zero frequency in one of the comparison groups.

**Table 2 bioengineering-11-01284-t002:** Detection Rates and Sex-Specific Distribution of the Top 20 Comorbidities in CAD Patients.

Comorbidities	Count (%)	Male (%)	Female (%)	OR (95% CI)
Hypertension	120 (61.54)	81 (58.70)	39 (68.42)	0.66 (0.35, 1.27)
Metabolic Diseases	43 (22.05)	35 (25.36)	8 (14.04)	2.00 (0.88, 4.54)
Dyslipidemia	41 (21.03)	23 (16.67)	18 (31.58)	0.43 (0.21, 0.88)
Diabetes and Its Complications	38 (19.49)	28 (20.29)	10 (17.54)	1.17 (0.53, 2.56)
Fatty Liver Disease	35 (17.95)	29 (21.01)	6 (10.53)	2.13 (0.86, 5.31)
Cardiac Arrhythmias	28 (14.36)	23 (16.67)	5 (8.77)	1.94 (0.73, 5.20)
Renal Cysts	23 (11.79)	21 (15.22)	2 (3.51)	4.06 (1.06, 15.64)
Post PCI	22 (11.28)	18 (13.04)	4 (7.02)	1.83 (0.62, 5.37)
Renal Dysfunction	18 (9.23)	13 (9.42)	5 (8.77)	1.03 (0.36, 2.91)
Pulmonary Infections	16 (8.21)	11 (7.97)	5 (8.77)	0.86 (0.30, 2.50)
Hematologic Diseases	15 (7.69)	8 (5.80)	7 (12.28)	0.44 (0.16, 1.23)
Gastrointestinal Diseases	15 (7.69)	9 (6.52)	6 (10.53)	0.58 (0.20, 1.66)
Atherosclerosis and Stenotic	15 (7.69)	11 (7.97)	4 (7.02)	1.07 (0.34, 3.34)
Cerebral Infarction	14 (7.18)	9 (6.52)	5 (8.77)	0.70 (0.23, 2.10)
Gallbladder Diseases	12 (6.15)	8 (5.80)	4 (7.02)	0.77 (0.24, 2.54)
Tobacco dependence	12 (6.15)	12 (8.70)	0 (0.00)	INF *
Myocardial Infarction	11 (5.64)	11 (7.97)	0 (0.00)	INF *
Liver Cystic Lesions	11 (5.64)	10 (7.25)	1 (1.75)	3.08 (0.54, 17.52)
Skeletal Diseases	10 (5.13)	5 (3.62)	5 (8.77)	0.39 (0.12, 1.34)
Valvular Heart Diseases	9 (4.62)	7 (5.07)	2 (3.51)	1.27 (0.29, 5.48)

* INF indicates cases where the odds ratio (OR) cannot be calculated due to a zero frequency in one of the comparison groups.

**Table 3 bioengineering-11-01284-t003:** Top Five Diagnoses of CAD by Age Group, Sex, and Admission Year.

Group *		Top1	Top2	Top3	Top4	Top5
Age	30–39	UA	Braunwald Class II B	Braunwald Class II B	/	/
	40–49	MI	KILLIP Class I	SA	Braunwald Class III B	CCS Angina Class II
	50–59	UA	Braunwald Class II B	MI	KILLIP Class I	CA
	60–69	UA	Braunwald Class II B	SA	CCS Angina Class II	Braunwald Class III B
	70–79	UA	SA	Braunwald Class II B	CCS Angina Class II	CA
	80–89	UA	SA	CCS Angina Class II	Braunwald Class II B	Braunwald Class III B
Sex	Male	UA	Braunwald Class II B	SA	MI	CCS Angina Class II
	Female	UA	Braunwald Class II B	SA	CCS Angina Class II	CA
Year	2013	UA	Braunwald Class II B	MI	SA	CA
	2014	SA	CCS Angina Class II	MI	UA	KILLIP Class I
	2015	UA	Braunwald Class II B	CCS Angina Class II	CA	Braunwald Class I B
	2016	UA	Braunwald Class II B	CCS Angina Class II	SA	ACS
	2017	SA	CCS Angina Class II	UA	MI	Braunwald Class II B
	2018	UA	Braunwald Class II B	SA	Braunwald Class III B	CA
	2019	UA	Braunwald Class II B	Braunwald Class III B	CA	SA
	2020	UA	Braunwald Class III B	MI	Braunwald Class II B	KILLIP Class I

* Please see Appendix A Table A1 for the abbreviation of diagnoses.

**Table 4 bioengineering-11-01284-t004:** Top Five Comorbidities of CAD by Age Group, Sex, and Admission Year.

Group *		Top1	Top2	Top3	Top4	Top5
Age	30–39	CI	Dyslipidemia	Fatty Liver Disease	Atherosclerosis	Post PCI
	40–49	Hypertension	Fatty Liver Disease	Dyslipidemia	Metabolic Diseases	Post PCI
	50–59	Hypertension	Arrhythmias	Fatty Liver Disease	Dyslipidemia	Metabolic Diseases
	60–69	Hypertension	Arrhythmias	Dyslipidemia	Metabolic Diseases	Diabetes
	70–79	Hypertension	Arrhythmias	Metabolic Diseases	Diabetes	Dyslipidemia
	80–89	Hypertension	Arrhythmias	Diabetes	Atherosclerosis	Metabolic Diseases
Sex	Male	Hypertension	Arrhythmias	Metabolic Diseases	Diabetes	Fatty Liver Disease
	Female	Hypertension	Dyslipidemia	Arrhythmias	Diabetes	Metabolic Diseases
Year	2013	Hypertension	Diabetes	Atherosclerosis	Dyslipidemia	Post PCI
	2014	Hypertension	Arrhythmias	Diabetes	Fatty Liver Disease	Metabolic Diseases
	2015	Hypertension	Diabetes	Atherosclerosis	Post PCI	Pulmonary Infections
	2016	Arrhythmias	Hypertension	Diabetes	Dyslipidemia	Metabolic Diseases
	2017	Hypertension	Arrhythmias	Atherosclerosis	CI	Diabetes
	2018	Hypertension	Metabolic Diseases	Dyslipidemia	Fatty Liver Disease	Arrhythmias
	2019	Hypertension	Dyslipidemia	Arrhythmias	Diabetes	Metabolic Diseases
	2020	Hypertension	Metabolic Diseases	Arrhythmias	Dyslipidemia	Fatty Liver Disease

* Please see Appendix A Table A2 for the abbreviation of comorbidities.

**Table 5 bioengineering-11-01284-t005:** Diagnosis–Comorbidity Pairs with Co-occurrence Frequency ≥ 10.

Diagnosis	Comorbidity	Co-Occurrence Frequency
Unstable Angina	Hypertension	59
Braunwald Class II B	Hypertension	31
Stable Angina	Hypertension	24
Unstable Angina	Dyslipidemia	20
CCS Angina class II	Hypertension	20
Unstable Angina	Metabolic Diseases	17
Myocardial Infarction	Hypertension	17
Unstable Angina	Fatty Liver Disease	16
Unstable Angina	Diabetes and Its Complications	14
Braunwald Class III B	Hypertension	14
Braunwald Class II B	Metabolic Diseases	12
Unstable Angina	Post PCI	12
Unstable Angina	Cardiac Arrhythmias	11
Braunwald Class II B	Dyslipidemia	10
Unstable Angina	Atherosclerosis and Stenotic	10

**Table 6 bioengineering-11-01284-t006:** Diagnoses with ≥10 Co-occurring Comorbidities in CAD Patients.

Diagnosis	Number of Associated Comorbidities	Diagnosis	Number of Associated Comorbidities
Unstable Angina	45	CCS Angina class III	16
Braunwald Class II B	40	NYHA Class III	15
Stable Angina	36	Coronary Atherosclerotic Heart Disease	14
Myocardial Infarction	32	CCS Angina class I	13
CCS Angina class II	31	Ischemic Cardiomyopathy	11
Braunwald Class III B	28	NYHA Class II	11
KILLIP Class I	23	Braunwald Class I B	10
Coronary Atherosclerosis	22	Multiple Vessel Disease	10

**Table 7 bioengineering-11-01284-t007:** Comorbidities with ≥10 Co-occurring Diagnoses in CAD Patients.

Comorbidity	Number of Associated Diagnoses	Comorbidity	Number of Associated Diagnoses
Hypertension	27	Renal Cysts	12
Metabolic Diseases	21	Hematologic Diseases	12
Diabetes and Its Complications	19	Fatty Liver Disease	11
Pulmonary Infections	19	Cerebral Infarction	11
Cardiac Arrhythmias	17	Gallbladder Diseases	11
Renal Dysfunction	16	Tobacco dependence	11
Dyslipidemia	15	Congenital Heart Diseases	11
Atherosclerosis and Stenotic	14	Gastrointestinal Diseases	10
Valvular Heart Diseases	13	Cardiac Diseases	10

## Data Availability

The raw data supporting the conclusions of this article are available from the corresponding authors upon reasonable request. In accordance with the Personal Information Protection Law of China (Article 4), anonymized data are not classified as personal information, and their use does not require additional ethical approval.

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
