# Peer review of "Population and Co-Occurrence Characteristics of Diagnoses and Comorbidities in Coronary Artery Disease Patients: A Case Study from a Hospital in Guangxi, China"

_bioengineering, 2024, doi:10.3390/bioengineering11121284_

Round 1
Reviewer 1 Report
Comments and Suggestions for Authors
I read the manuscript with great interest. The topic as a whole is very interesting and deserves to be published. However, the manuscript has several very serious limitations that raise serious doubts. The study includes patients who were hospitalized over an 8-year period with a diagnosis of coronary artery disease. Coronary artery disease is a very common diagnosis. Therefore, it is highly unlikely that the diagnosis was recorded in an average of only 24 patients each year. This raises doubts about the validity of the data recording in the hospital database and hence may suggest a significant selection bias. Even regardless of the low number of patients each year, the sample size as a whole is very small and makes network analysis difficult. The methods section is long, but it is not detailed in the important points (for example, there is no specific details on the models used) and there is no citation of the articles on which the models are based (for example, the diagnosis rate). It is not clear why the researchers focused on comparing genders from the beginning. This does not follow the purpose of the study.
The choice of diagnoses is unclear and too broad given the small number of people included in the study.
To summarize, the manuscript is interesting but raises questions about the validity of the data (only one patient with heart failure?).
Author Response
|
Comments 1: I read the manuscript with great interest. The topic as a whole is very interesting and deserves to be published. However, the manuscript has several very serious limitations that raise serious doubts. The study includes patients who were hospitalized over an 8-year period with a diagnosis of coronary artery disease. Coronary artery disease is a very common diagnosis. Therefore, it is highly unlikely that the diagnosis was recorded in an average of only 24 patients each year. This raises doubts about the validity of the data recording in the hospital database and hence may suggest a significant selection bias. |
|
Response 1: Thank you for your interest in our manuscript and for highlighting this important concern. We agree that the small number of patients each year appears unusual for a common diagnosis like coronary artery disease (CAD). To address this, we have added a detailed explanation regarding the data extraction process and inclusion criteria in the revised manuscript (Page 3, Paragraph 1, Lines 104-107). “[The 195 CAD patients with full follow-up information have been diagnosed according to authoritative professional guide [34-36], and CAD patients without full follow-up in-formation were not included in this dataset.]” Specifically, we clarified that the study focused exclusively on patients who met strict inclusion criteria, such as those with full follow-up information. This approach inevitably limited the sample size. Additionally, we have acknowledged the potential for selection bias in the "Limitations" section of the Discussion (Page 15, Paragraph 3, Lines 599-605). “[The dataset, derived from a single hospital in Guangxi, may not fully represent the broader population. With a sample size of 195 patients who had complete follow-up information, some degree of sampling error is expected. The low number of patients with certain diagnoses, such as heart failure, reflects the strict inclusion criteria, which are directly related to the sample size. Future studies should aim to include multi-center data to improve the generalizability of the findings.]” |
|
Comments 2: Even regardless of the low number of patients each year, the sample size as a whole is very small and makes network analysis difficult. |
|
Response 2: We acknowledge the reviewer’s concern regarding the sample size and its impact on the robustness of the network analysis. In response, we have elaborated on the methodology to justify why the network analysis was still applicable despite the small sample size. Specifically, we employed techniques which have been cited in the revised manuscript (Page 4, Paragraph 4, Lines 158-162, 171-174). “[we used the preprocessed diagnosis and comorbidity descriptions to construct two types of weighted co-occurrence networks [38, 39], incorporating both homogeneous and heterogeneous nodes [40]. The first type is a monopartite graph consisting of nodes of the same type (diagnosis or comorbidity) and the second type is a bipartite graph consisting of nodes of different types (diagnosis and comorbidity).]” “[While the sample size in this study is relatively small, we implemented several strategies to ensure the robustness of the network analysis, including the use of weighted networks to enhance reliability, and cross-validation with existing literature. These steps mitigate the limitations associated with small datasets and support the validity of the findings.]” Additionally, we have performed sensitivity analyses (Page 14-15, Section 3.6. Sensitivity Analysis of Co-occurrence Networks, Lines 458-511) to evaluate the reliability of the network metrics and included the results in the Supplementary Material (Page 19-21, Appendix Table A3/ Figure A1-A3, Lines 697-711). |
|
Comments 3: The methods section is long, but it is not detailed in the important points (for example, there is no specific details on the models used) and there is no citation of the articles on which the models are based (for example, the diagnosis rate). |
|
Response 3: Thank you for pointing this out. We have revised the Methods section to include specific details about the models used and the references that informed our approach (Page 3-5, Lines 98-212). In particular, we have now provided: 1. A detailed description of the network construction process, including the algorithms and statistical models applied. 2. Citations to relevant studies that support our methodology, including the calculation of diagnosis rates and network metrics. These additions not only clarify our methodology but also ensure that our work is grounded in established research. Update in Section 2.2 “[The 195 CAD patients with full follow-up information have been diagnosed according to authoritative professional guide [34-36], and CAD patients without full follow-up in-formation were not included in this dataset.]” Update in Section 2.3 “[The diagnosis and comorbidity information for each patient was complete. From the original text records of diagnoses and comorbidities, we initially extracted 57 distinct diagnosis descriptions and 272 comorbidity descriptions. Due to occasional typographical errors in the original records, we merged similar descriptions of diagnoses and comorbidities based on clinicians' expert opinions. After this preprocessing step, we obtained 32 unique diagnosis descriptions and 52 comorbidity-related descriptions.]” Update in Section 2.4 “[The detection rate (DR) for each diagnosis and comorbidity was calculated as the ratio of the number of specific diagnoses or comorbidities to the total number of CAD-related records, following the approach outlined in Reference [37], expressed as a percentage:]” Update in Section 2.5 “[The four metrics—average clustering coefficient [43], modularity [44], description length [45, 46] and betweenness centrality [42, 47]—were used to offer complementary insights into the structural and functional properties of the co-occurrence networks. The clustering coefficient emphasizes local cohesiveness, modularity evaluates global community structure, and betweenness centrality identifies key nodes for connectivity and flow. Together, they provide a comprehensive framework for network analysis.]” |
|
Comments 4: It is not clear why the researchers focused on comparing genders from the beginning. This does not follow the purpose of the study. |
|
Response 4: We appreciate the reviewer’s observation and recognize the need to clarify the rationale for the focus on gender comparisons. In order to make the research content, title and purpose consistent, we further emphasized the necessity of studying the population characteristics of the diagnosis and comorbidity in the introduction, and modified the title of the paper to enhance the readability of the article. The new title is “Population and Co-occurrence Characteristics of Diagnoses and Comorbidities in Coronary Artery Disease Patients: A Case Study from a Hospital in Guangxi, China”, which maintains academic rigor while clearly indicating the regional and institutional context of the study. In the revised manuscript, we have explicitly stated that the gender-based analysis was motivated by well-documented differences in CAD presentation, progression, and outcomes between males and females. This focus aligns with the study’s aim of identifying patterns that can inform personalized management strategies. The rationale has been added to the Introduction (Page 1, Paragraph 1, Lines 39-42). “[Gender and age differences in CAD presentation and outcomes are well-documented in the literature [9-15] and were a key motivation for this analysis. Understanding these differences is critical for developing tailored management strategies.]” |
|
Comments 5: The choice of diagnoses is unclear and too broad given the small number of people included in the study. |
|
Response 5: Thank you for raising this issue. To address this concern, we have provided a clearer explanation of the criteria used to select the diagnoses included in the study. The diagnoses included in this study were identified based on AHA guidelines (References 34-36). This information has been added to the Methods section (Page 3, Paragraph 1, Lines 104-107). “[The 195 CAD patients with full follow-up information have been diagnosed according to authoritative professional guide [34-36], and CAD patients without full follow-up in-formation were not included in this dataset.]” |
|
Comments 6: To summarize, the manuscript is interesting but raises questions about the validity of the data (only one patient with heart failure?). |
|
Response 6: We appreciate the reviewer’s comments and agree that the low number of patients with certain diagnoses, such as heart failure, may appear unusual. To address this, we have reviewed the data and clarified in the revised manuscript that the strict inclusion criteria. The low number of patients with certain diagnoses, such as heart failure, reflects the strict inclusion criteria, which are directly related to the sample size. With a sample size of 195 patients who had complete follow-up information, some degree of sampling error is expected. However, in terms of proportion (1/195), the incidence of heart failure is acceptable. This is because patients with complete follow-up information were unlikely to have chronic heart failure, thereby making the likelihood of acute heart failure relatively low. This explanation has been added to the Methods section (Page 3, Paragraph 1, Lines 104-107) and the Discussion (Page 16, Paragraph 4, Lines 601-604). “[The 195 CAD patients with full follow-up information have been diagnosed according to authoritative professional guide [34-36], and CAD patients without full follow-up in-formation were not included in this dataset.]” “[With a sample size of 195 patients who had complete follow-up information, some degree of sampling error is expected. The low number of patients with certain diagnoses, such as heart failure, reflects the strict inclusion criteria, which are directly related to the sample size.]” |
Reviewer 2 Report
Comments and Suggestions for Authors
1. Sample Size and Representativeness:
• The dataset includes only 195 patients from a single hospital in Guangxi, China. This limits the generalizability of findings to other populations and regions.
• The study could benefit from multi-center data collection to strengthen its applicability.
2. Retrospective Design:
• A reliance on retrospective data introduces potential biases due to incomplete or inconsistent medical records.
• This limitation makes it difficult to draw causal inferences or account for longitudinal dynamics in comorbidities.
3. Lack of Longitudinal Analysis:
• The study focuses on static co-occurrence patterns but does not examine how comorbidities evolve over time.
• Incorporating longitudinal data could provide insights into disease progression and temporal relationships.
4. Data Normalization Challenges:
• While the use of NLP and Python scripts for data preprocessing is noted, the reliance on manual standardization may introduce errors or inconsistencies.
• Automation and validation methods should be further emphasized for accuracy.
5. Geographical Variability:
• The findings may not account for regional variations in risk factors, healthcare access, or socioeconomic conditions.
• Broader geographic sampling would enhance the study’s robustness.
6. Visual and Statistical Analysis Limitations:
• The network visualizations and modularity analysis provide valuable insights but could be more detailed.
• Including sensitivity analyses or bootstrapping methods might strengthen the reliability of the network metrics.
7. Discussion and Implications:
• The discussion identifies hypertension and metabolic disorders as central hubs but could delve deeper into specific intervention strategies tailored to these findings.
• The implications for personalized medicine and integrated care are not fully explored.
8. Ethical and Data Concerns:
• Although anonymization is mentioned, there is limited detail on how patient confidentiality and ethical considerations were upheld.
• Clearer documentation of ethical approvals and consent processes would improve transparency.
Author Response
|
Comments 1: Sample Size and Representativeness: l The dataset includes only 195 patients from a single hospital in Guangxi, China. This limits the generalizability of findings to other populations and regions. l The study could benefit from multi-center data collection to strengthen its applicability. |
|
Response 1: Thank you for highlighting the limitations of our sample size and representativeness. We agree that the use of data from a single hospital may limit the generalizability of our findings. To address this, we have added a detailed discussion of this limitation in the revised manuscript (Page 16, Paragraph 4, Lines 599-605). Specifically, we acknowledge the need for future studies to incorporate multi-center data collection to enhance external validity. “[The dataset, derived from a single hospital in Guangxi, may not fully represent the broader population. With a sample size of 195 patients who had complete follow-up information, some degree of sampling error is expected. The low number of patients with certain diagnoses, such as heart failure, reflects the strict inclusion criteria, which are directly related to the sample size. Future studies should aim to include multi-center data to improve the generalizability of the findings.]” |
|
Comments 2: Retrospective Design: l A reliance on retrospective data introduces potential biases due to incomplete or inconsistent medical records. l This limitation makes it difficult to draw causal inferences or account for longitudinal dynamics in comorbidities. |
|
Response 2: We appreciate the reviewer’s observation regarding the inherent limitations of a retrospective design. We have acknowledged the potential biases due to incomplete or inconsistent medical records and clarified that the study’s cross-sectional nature precludes establishing causal relationships. We have now explicitly addressed this concern in the revised manuscript (Page 16, Paragraph 4, Lines 608-627). “[The retrospective design of the study may introduce biases arising from incomplete or inconsistent medical records. Future studies with larger, more diverse cohorts and prospective designs are essential to validate these findings and explore the underlying mechanisms driving the observed sex- and age-specific patterns in CAD diagnoses and comorbidities. Moreover, while the study identified significant associations between certain diagnoses and comorbidities, the cross-sectional nature of the analysis precludes the establishment of causal relationships. The focus of this study on static co-occurrence patterns provides valuable insights but does not capture the temporal dynamics of disease progression. Future prospective studies are needed to address these limitations. Incorporating longitudinal data in future studies could reveal the evolution of comorbidities and their causal relationships, offering more targeted intervention strategies.]” |
|
Comments 3: Lack of Longitudinal Analysis: l The study focuses on static co-occurrence patterns but does not examine how comorbidities evolve over time. l Incorporating longitudinal data could provide insights into disease progression and temporal relationships. |
|
Response 3: Thank you for raising this important point. Although Tables 3 and 4 in this study show the changes in the top 5 diagnoses and comorbidities, the dynamic evolution rules and mechanisms of the association between diagnoses and comorbidities still need further in-depth research. We agree that longitudinal data would provide deeper insights into the progression of comorbidities and their temporal relationships. In the revised manuscript, we have included this as a direction for future research in the Discussion section (Page 16, Paragraph 4, Lines 615-627). “[The focus of this study on static co-occurrence patterns provides valuable insights but does not capture the temporal dynamics of disease progression. Incorporating longitudinal data in future studies could reveal the evolution of comorbidities and their causal relationships, offering more targeted intervention strategies.]” |
|
Comments 4: Data Normalization Challenges: l While the use of NLP and Python scripts for data preprocessing is noted, the reliance on manual standardization may introduce errors or inconsistencies. l Automation and validation methods should be further emphasized for accuracy. |
|
Response 4: We appreciate the reviewer’s concern regarding data normalization. To address this, we have expanded on the data preprocessing steps in the Methods section, specifying the use of both automated and manual processes (Page 3, Paragraph 2, Lines 117-119). “[The data normalization and standardization process has been confirmed and verified with professional clinical doctors, which involved several steps to ensure consistency and accuracy.]” We have also acknowledged the potential for manual standardization errors and emphasized the importance of validation methods (Page 3-4, Lines 131-139). “[The diagnosis and comorbidity information for each patient was complete. From the original text records of diagnoses and comorbidities, we initially extracted 57 distinct diagnosis descriptions and 272 comorbidity descriptions. Due to occasional typographical errors in the original records, we merged similar descriptions of diagnoses and comorbidities based on clinicians' expert opinions. After this preprocessing step, we obtained 32 unique diagnosis descriptions and 52 comorbidity-related descriptions.]” |
|
Comments 5: Geographical Variability: l The findings may not account for regional variations in risk factors, healthcare access, or socioeconomic conditions. l Broader geographic sampling would enhance the study’s robustness. |
|
Response 5: Thank you for pointing out the potential impact of geographical variability on our findings. We agree that broader geographic sampling would provide a more comprehensive understanding of regional differences in CAD risk factors and management. We have now acknowledged this limitation in the revised manuscript (Page 17, Paragraph 1, Lines 629-633). “[The findings of this study are based on data from a single region and may not account for geographical variations in risk factors, healthcare access, or socioeconomic conditions. Future studies should include data from diverse geographic regions to enhance the robustness of the conclusions.]” |
|
Comments 6: Visual and Statistical Analysis Limitations: l The network visualizations and modularity analysis provide valuable insights but could be more detailed. l Including sensitivity analyses or bootstrapping methods might strengthen the reliability of the network metrics. |
|
Response 6: We thank the reviewer for this valuable suggestion. To address this, we have enhanced the description of the network analysis in the Methods section (Page 4-5, Lines 181-204, 210-212). “[The four metrics—average clustering coefficient [43], modularity [44], description length [45, 46] and betweenness centrality [42, 47]—were used to offer complementary insights into the structural and functional properties of the co-occurrence networks. The clustering coefficient emphasizes local cohesiveness, modularity evaluates global community structure, and betweenness centrality identifies key nodes for connectivity and flow. Together, they provide a comprehensive framework for network analysis. …To strengthen the robustness of the network metrics, sensitivity analyses based on adjusting edge weight filtering thresholds were conducted, validating the reliability of the network structure and modularity findings.]” Additionally, we have performed sensitivity analyses (Page 14-15, Section 3.6. Sensitivity Analysis of Co-occurrence Networks, Lines 458-498) to evaluate the reliability of the network metrics and included the results in the Supplementary Material (Page 19-21, Appendix Table A3/ Figure A1-A3, Lines 697-711). |
|
Comments 7: Discussion and Implications: l The discussion identifies hypertension and metabolic disorders as central hubs but could delve deeper into specific intervention strategies tailored to these findings. l The implications for personalized medicine and integrated care are not fully explored. |
|
Response 7: We appreciate the reviewer’s suggestion to expand the discussion on intervention strategies and implications for personalized medicine. In the revised Discussion section (Page 16, Paragraph 1, Lines 565-569), we have provided additional details on potential intervention strategies for hypertension and metabolic disorders. We have also elaborated on how the findings could inform personalized management and integrated care approaches. “[The identification of hypertension and metabolic disorders as central hubs highlights opportunities for targeted interventions, such as aggressive blood pressure control and metabolic risk management. These findings also underscore the importance of integrated care models that address the multifaceted nature of CAD comorbidities, paving the way for personalized treatment plans.]” |
|
Comments 8: Ethical and Data Concerns: l Although anonymization is mentioned, there is limited detail on how patient confidentiality and ethical considerations were upheld. l Clearer documentation of ethical approvals and consent processes would improve transparency. |
|
Response 8: We agree with the reviewer’s comment and have added further details on the ethical considerations in Section “Institutional Review Board Statement” and Section “Data Availability Statement” (Page 17, Lines 665-675). Specifically, we have clarified how patient confidentiality was maintained and provided the ethical approval reference number and details on the consent processes. "[Institutional Review Board Statement: This study was conducted in accordance with strict ethical guidelines to ensure the protection of patient confidentiality. All patient data were fully anonymized prior to analysis. Ethical approval for the study was obtained from the Ethics Committee of Guangxi Zhuang Autonomous Region People's Hospital (Approval Number: KY-KJT-2021-95). The retrospective nature of the data collection complies with institutional and national regulations, and the requirement for formal patient consent was waived by the ethics committee.]" “[Data Availability Statement: The raw data supporting the conclusions of this article are available from the corresponding authors upon reasonable request. In accordance with the Personal Information Protection Law of China (Article 4), anonymized data are not classified as personal in-formation, and their use does not require additional ethical approval.]” |
Reviewer 3 Report
Comments and Suggestions for Authors
In this study, network analysis was utilized to investigate the patterns of diagnoses and comorbidities in patients with coronary artery disease (CAD). A retrospective analysis was conducted on 195 patients treated at a hospital in Guangxi, China. The study identified significant differences based on age and gender, highlighting the frequent association of hypertension and metabolic disorders with CAD. Network analysis revealed central nodes such as hypertension and unstable angina, and these findings provide insights for developing personalized treatment strategies. By focusing on patient data from the Guangxi region, the study offers valuable information for local health policies. Furthermore, by examining age- and gender-specific patterns in CAD diagnoses and comorbidities, it holds potential to propose more targeted treatment approaches. However, there are several limitations to consider:
- The study is limited to a sample size of only 195 patients, which is insufficient for generalization.
- The data were collected from a single hospital, restricting geographic and demographic diversity. This narrow scope significantly limits the analysis, and studies of this nature are more suitable for presentation at conferences.
- Due to the retrospective nature of the data, it is challenging to assess the temporal progression of diagnoses and comorbidities.
- The manual consolidation of definitions during data normalization carries a risk of bias, which is critical for such data analyses.
- The mathematical and statistical foundations do not provide robust answers due to the limitations of the dataset.
- From an application perspective, while the study offers ideas for integrated treatment approaches, its findings need validation in larger populations.
- I would kindly suggest updating the references used in the study to include more recent sources.
Finally, regarding the discussion section, I believe that the study's contribution to the existing literature is limited. There are already numerous studies in this field, and this work does not appear to significantly advance the current body of knowledge.
Author Response
|
Comments 1: The study is limited to a sample size of only 195 patients, which is insufficient for generalization. |
|
Response 1: Thank you for pointing this out. We agree with this comment. Therefore, we have added a discussion on the limitations of our sample size and its implications for the generalizability of our findings. We have also included a section on future research where we suggest the need for studies with larger and more diverse samples to validate our results. This change can be found on Page 16, line 599-605. “[The dataset, derived from a single hospital in Guangxi, may not fully represent the broader population. With a sample size of 195 patients who had complete follow-up information, some degree of sampling error is expected. The low number of patients with certain diagnoses, such as heart failure, reflects the strict inclusion criteria, which are directly related to the sample size. Future studies should aim to include multi-center data to improve the generalizability of the findings.]” |
|
Comments 2: The data were collected from a single hospital, restricting geographic and demographic diversity. This narrow scope significantly limits the analysis, and studies of this nature are more suitable for presentation at conferences. |
|
Response 2: We acknowledge the reviewer's concern regarding the geographic and demographic limitations of our data. To address this, we have added a section in the discussion that highlights the need for multi-center studies to enhance the diversity of the data and to better understand the broader implications of our findings. We have also noted the limitations of our single-center study in the context of the current literature. This change can be found on Page 16-17, line 605-612. “[Data preprocessing involved a combination of automated NLP techniques and manual standardization. While this approach ensured contextual accuracy, it may have introduced inconsistencies. Future studies should incorporate fully automated and validated workflows to enhance reproducibility and accuracy. The retrospective design of the study may introduce biases arising from incomplete or inconsistent medical records. Future studies with larger, more diverse cohorts and prospective designs are essential to validate these findings and explore the underlying mechanisms driving the observed sex- and age-specific patterns in CAD diagnoses and comorbidities.]” |
|
Comments 3: Due to the retrospective nature of the data, it is challenging to assess the temporal progression of diagnoses and comorbidities. |
|
Response 3: We agree with the reviewer's observation about the limitations of retrospective data. We have revised the manuscript to include a discussion on the challenges associated with retrospective data analysis and how this might affect the interpretation of the temporal progression of diagnoses and comorbidities. We have also emphasized the importance of prospective studies in future research to overcome these limitations. This change can be found on Page 16-17, line 612-633. “[Moreover, while the study identified significant associations between certain diagnoses and comorbidities, the cross-sectional nature of the analysis precludes the establishment of causal relationships. The focus of this study on static co-occurrence patterns provides valuable insights but does not capture the temporal dynamics of disease progression. Future prospective studies are needed to address these limitations. Incorporating longitudinal data in future studies could reveal the evolution of comorbidities and their causal relationships, offering more targeted intervention strategies. Longitudinal studies are necessary to elucidate the temporal dynamics of diagnoses and comorbid conditions in CAD patients, thereby informing more effective prevention and intervention strategies. The findings of this study are based on data from a single region and may not account for geographical variations in risk factors, healthcare access, or socioeconomic conditions. Future studies should include data from diverse geographic regions to enhance the robustness of the conclusions.]” |
|
Comments 4: The manual consolidation of definitions during data normalization carries a risk of bias, which is critical for such data analyses. |
|
Response 4: Thank you for highlighting this potential source of bias. We have taken steps to minimize bias in our data normalization process by implementing a standardized protocol for definition consolidation. We have also added a section in the methods that details our protocol and discusses the measures taken to reduce bias. This change can be found on Page 3-4, line 117-139. “[The data normalization and standardization process has been confirmed and verified with professional clinical doctors, which involved several steps to ensure consistency and accuracy. First, manual resolution of textual ambiguities and synonyms was performed; for example, diagnostic entries such as "triple vessel disease," "left main disease," "triple coronary disease," and "multivessel disease" were consolidated under the standardized term "coronary multivessel disease." Similarly, various descriptions of hypertension, including "hypertension stage 3 very high-risk group," "hypertension stage 2 high-risk group," and others, were unified under the general term "hypertension." Additionally, typographical errors and inconsistencies in the original records were corrected, and duplicate entries were removed through deduplication preprocessing. These steps aimed to ensure the accuracy and consistency of diagnosis and comorbidity classifications while preserving the integrity of the original patient records. Ultimately, all medical diagnostic and comorbidity information was converted to standardized terms, facilitating streamlined manipulation and analysis. The diagnosis and comorbidity information for each patient was complete. From the original text records of diagnoses and comorbidities, we initially extracted 57 distinct diagnosis descriptions and 272 comorbidity descriptions. Due to occasional typographical errors in the original records, we merged similar descriptions of diagnoses and comorbidities based on clinicians' expert opinions. After this preprocessing step, we obtained 32 unique diagnosis descriptions and 52 comorbidity-related descriptions.]” |
|
Comments 5: The mathematical and statistical foundations do not provide robust answers due to the limitations of the dataset. |
|
Response 5: We sincerely appreciate the reviewer’s comments regarding the limitations of the dataset and the statistical results. Below, we address these concerns and highlight the significance of our study: 1) Acknowledgment of Dataset Limitations and Statistical Results: We acknowledge that the statistical significance of certain results, such as the odds ratio (OR) calculations, is affected by the limitations of the dataset, including sample size and data distribution. These constraints may limit the robustness of some conclusions and are inherent to the available data. However, we believe these results still provide valuable insights into the relationships and patterns within the dataset, serving as a foundation for future studies. 2) Sensitivity Analysis of Network Structure: To address concerns about robustness, we have conducted a network structure sensitivity analysis (Page 14-15, Section “3.6. Sensitivity Analysis of Co-occurrence Networks”, Lines 458-511) and included the results in the Supplementary Material (Page 19-21, Appendix Table A3/ Figure A1-A3, Lines 697-711). This includes testing the stability of the co-occurrence network under different edge weight filtering thresholds and evaluating the consistency of core nodes and edges. The results demonstrate that the key relationships identified in the study, such as the central roles of Unstable Angina, Hypertension, and other major comorbidities, remain stable across different filtering thresholds. This analysis strengthens the reliability of the network findings despite the statistical limitations of individual metrics. 3) Significance of the Study: (1) Novel Insights into Coronary Artery Disease (CAD): Despite the dataset limitations, this study provides a novel and comprehensive view of the co-occurrence patterns between CAD diagnoses and comorbidities. By integrating network analysis, the study identifies central conditions (e.g., Unstable Angina and Hypertension) and their key connections, which highlight potential targets for clinical management and further investigation. (2) Methodological Value: The approach used in this study, combining co-occurrence networks with sensitivity analyses, offers a replicable framework for analyzing complex relationships in medical datasets. This methodology can be applied to other diseases or datasets, making it a valuable reference for researchers in related fields. 4) Open Data for Future Research: To further enhance the impact of this study, we are committed to making the dataset available to researchers in related fields. By providing access to the anonymized dataset, we aim to promote transparency, reproducibility, and collaboration. We believe this will enable other researchers to validate our findings, explore alternative methodologies, and generate new insights into CAD and its comorbidities. 5) Broader Impact and Future Directions: While certain statistical results may lack significance due to dataset limitations, the study serves as a proof of concept for using network analysis to reveal clinically relevant patterns. Future studies based on larger and more diverse datasets could further validate and expand upon these findings. Additionally, the study highlights the importance of collaborative efforts to improve data availability and quality in the field of CAD research. |
|
Comments 6: From an application perspective, while the study offers ideas for integrated treatment approaches, its findings need validation in larger populations. |
|
Response 6: We agree that our findings require validation in larger populations. We have added a section in the discussion that outlines the need for future studies to validate our findings in larger and more diverse populations. We have also emphasized the preliminary nature of our results and the importance of further research for clinical application. This change can be found on Page 16-17, line 610-633. “[Future studies with larger, more diverse cohorts and prospective designs are essential to validate these findings and explore the underlying mechanisms driving the observed sex- and age-specific patterns in CAD diagnoses and comorbidities. Moreover, while the study identified significant associations between certain diagnoses and comorbidities, the cross-sectional nature of the analysis precludes the establishment of causal relationships. The focus of this study on static co-occurrence patterns provides valuable insights but does not capture the temporal dynamics of disease progression. Future prospective studies are needed to address these limitations. Incorporating longitudinal data in future studies could reveal the evolution of comorbidities and their causal relationships, offering more targeted intervention strategies. Longitudinal studies are necessary to elucidate the temporal dynamics of diagnoses and comorbid conditions in CAD patients, thereby informing more effective prevention and intervention strategies. The findings of this study are based on data from a single region and may not account for geographical variations in risk factors, healthcare access, or socioeconomic conditions. Future studies should include data from diverse geographic regions to enhance the robustness of the conclusions.]” |
|
Comments 7: I would kindly suggest updating the references used in the study to include more recent sources. |
|
Response 7: We appreciate the suggestion to update our references. We have conducted a literature review to identify and include more recent and relevant studies in our reference list. This will provide a more current context for our work and enhance the relevance of our findings. This change can be found throughout the manuscript, with the updated reference list in Section “Reference” on Page 21-24. Now the total number of literatures is increased to 63. |
|
Comments 8: Regarding the discussion section, I believe that the study's contribution to the existing literature is limited. There are already numerous studies in this field, and this work does not appear to significantly advance the current body of knowledge. |
|
Response 8: We understand the reviewer's concern about the contribution of our study to the existing literature. We have revised the discussion section to more clearly articulate how our findings, despite the limitations, offer novel insights into the patterns of diagnoses and comorbidities in CAD patients from the Guangxi region. We have also highlighted the potential for our study to inform local health policies and the need for further research to expand upon our preliminary findings. This change can be found on page 16, paragraph 2-3, line 579-603 and page 17, line 643-649. “[While our findings align with previous research, the network analysis approach provides unique insights by identifying central hubs of comorbidities and their interactions. This novel perspective enables a more targeted and integrated approach to managing coro-nary artery disease, advancing both clinical and epidemiological practices. Furthermore, the bipartite co-occurrence network illustrated complex interactions between diagnoses and comorbidities, reinforcing the necessity for holistic treatment plans that consider the interplay of multiple health conditions in CAD patients. The edge weight filtering strategy plays a crucial role in refining network structures by removing low-weight edges that may introduce noise or obscure significant connections. This process highlights the core relationships within networks, resulting in increased modularity and reduced network complexity as reflected by lower description lengths. By filtering out weaker connections, co-occurrence networks become more interpretable, with clearer community structures and enhanced modular characteristics. However, this comes at the cost of network sparsity, as filtering reduces edge density, average degree, and clustering coefficients, potentially affecting the overall connectivity and local clustering properties of the network. For practical applications, moderate edge weight filtering is recommended for diagnosis co-occurrence networks to enhance the identification of significant diagnostic patterns while retaining sufficient connectivity. In comorbidity co-occurrence networks, higher thresholds can effectively capture key comorbidities without being influenced by weaker associations. In diagnosis-comorbidity bipartite networks, filtering strategies should balance sparsity and interpretability, focusing on extracting critical diagnostic-comorbidity relationships. These approaches provide valuable insights into the core structures of the networks, enabling more robust and clinically meaningful analyses of disease patterns and associations.]” “[Our study advances the understanding of coronary artery disease by leveraging network analysis to uncover novel co-occurrence patterns and central hubs, such as the critical roles of Unstable Angina and Hypertension in CAD comorbidity networks. These findings go beyond reaffirming existing literature by providing actionable insights into prioritizing comorbidity management and identifying targets for early intervention strategies, offering a deeper understanding of the complex interplay between diagnoses and comorbidities.]” |
Reviewer 4 Report
Comments and Suggestions for Authors
The authors Wang et al present an intriguing study that investigates the role of network analysis of co-occurring diagnoses and comorbidities in coronary artery disease as a tool for personalized patient management. While the study is compelling, significant revisions and corrections are required before determining its suitability for publication.
For instance, in the Materials and Methods section (2.1 Experimental Design), the authors reference Figure 1 as a comprehensive flowchart outlining the experimental design. However, the actual Figure 1 in the manuscript depicts the age distribution of hospitalized coronary artery disease patients. This inconsistency makes it difficult to discern their exact methodology. The authors should ensure the figures are correctly labeled and aligned with the content.
Additionally, I recommend improving the manuscript's readability by eliminating redundancies. For example, the authors state the number of patients included in the study in both Section 2.2 and Section 2.4, which is unnecessary and repetitive.
The results are presented with appropriate detail, but the figures referenced in the results section require significant improvement. While figures can be excellent visual aids, the quality of Figures 2 and 3 is poor; even at maximum zoom, the text is nearly illegible. The same issue applies to Figures 4 and 5. I strongly encourage the authors to provide higher-quality versions of these figures to enhance their clarity and usefulness.
In the discussion section, the authors should expand on how their findings contribute to the clinical domain. They acknowledge that many results from their network analysis align with previous clinical research. This raises the question: what unique insights or advantages does their novel network analysis approach bring to the clinical and epidemiological fields? Highlighting these contributions would strengthen the discussion.
Finally, in the conclusion, the authors claim that their study enhances the understanding of the complex interplay between various diagnoses and coronary artery disease. However, the discussion largely suggests that their findings reaffirm existing literature. The authors should explicitly detail how their findings advance or deepen the current understanding of these relationships to justify their claims.
Author Response
|
Comments 1: The authors Wang et al present an intriguing study that investigates the role of network analysis of co-occurring diagnoses and comorbidities in coronary artery disease as a tool for personalized patient management. While the study is compelling, significant revisions and corrections are required before determining its suitability for publication. For instance, in the Materials and Methods section (2.1 Experimental Design), the authors reference Figure 1 as a comprehensive flowchart outlining the experimental design. However, the actual Figure 1 in the manuscript depicts the age distribution of hospitalized coronary artery disease patients. This inconsistency makes it difficult to discern their exact methodology. The authors should ensure the figures are correctly labeled and aligned with the content. |
|
Response 1: Thank you for pointing this out. We agree with this comment. Therefore, we have revised the manuscript to correct the inconsistency between the referenced Figure 1 and its content. We have replaced the incorrect Figure 1 with a new figure that accurately represents the experimental design as described in the Materials and Methods section. This change can be found on Page 3, line 98-99. |
|
Comments 2: Additionally, I recommend improving the manuscript's readability by eliminating redundancies. For example, the authors state the number of patients included in the study in both Section 2.2 and Section 2.4, which is unnecessary and repetitive. |
|
Response 2: Agree. We have, accordingly, revised the manuscript to remove redundancies and improve readability. Specifically, we have removed the duplicate statement regarding the number of patients included in the study from Section 2.4, as it was already mentioned in Section 2.2. |
|
Comments 3: The results are presented with appropriate detail, but the figures referenced in the results section require significant improvement. While figures can be excellent visual aids, the quality of Figures 2 and 3 is poor; even at maximum zoom, the text is nearly illegible. The same issue applies to Figures 4 and 5. I strongly encourage the authors to provide higher-quality versions of these figures to enhance their clarity and usefulness. |
|
Response 3: Thank you for this feedback. We have taken your comments to heart and have provided higher-quality versions of Figures 2, 3, 4, and 5. We have enhanced the resolution and clarity of these figures to ensure that all text and details are legible and useful for the reader. These improved figures are renumbered Figures 3, 4, 5, and 6 which can be found on Page 6, 12, and 13, corresponding to the results section. |
|
Comments 4: In the discussion section, the authors should expand on how their findings contribute to the clinical domain. They acknowledge that many results from their network analysis align with previous clinical research. This raises the question: what unique insights or advantages does their novel network analysis approach bring to the clinical and epidemiological fields? Highlighting these contributions would strengthen the discussion. |
|
Response 4: We appreciate this suggestion and have expanded the discussion section to better articulate how our findings contribute to the clinical domain. We have added a new paragraph that discusses the unique insights and advantages of our novel network analysis approach, which were not previously highlighted in the clinical and epidemiological literature. This addition can be found on Page 16, line 579-603. “[While our findings align with previous research, the network analysis approach provides unique insights by identifying central hubs of comorbidities and their interactions. This novel perspective enables a more targeted and integrated approach to managing coronary artery disease, advancing both clinical and epidemiological practices. Furthermore, the bipartite co-occurrence network illustrated complex interactions between diagnoses and comorbidities, reinforcing the necessity for holistic treatment plans that consider the interplay of multiple health conditions in CAD patients. The edge weight filtering strategy plays a crucial role in refining network structures by removing low-weight edges that may introduce noise or obscure significant connections. This process highlights the core relationships within networks, resulting in in-creased modularity and reduced network complexity as reflected by lower description lengths. By filtering out weaker connections, co-occurrence networks become more interpretable, with clearer community structures and enhanced modular characteristics. However, this comes at the cost of network sparsity, as filtering reduces edge density, average degree, and clustering coefficients, potentially affecting the overall connectivity and local clustering properties of the network. For practical applications, moderate edge weight filtering is recommended for diagnosis co-occurrence networks to enhance the identification of significant diagnostic patterns while retaining sufficient connectivity. In comorbidity co-occurrence networks, higher thresholds can effectively capture key comorbidities without being influenced by weaker associations. In diagnosis-comorbidity bipartite networks, filtering strategies should balance sparsity and interpretability, focusing on extracting critical diagnostic-comorbidity relationships. These approaches provide valuable insights into the core structures of the networks, enabling more robust and clinically meaningful analyses of disease patterns and associations.]” |
|
Comments 5: Finally, in the conclusion, the authors claim that their study enhances the understanding of the complex interplay between various diagnoses and coronary artery disease. However, the discussion largely suggests that their findings reaffirm existing literature. The authors should explicitly detail how their findings advance or deepen the current understanding of these relationships to justify their claims. |
|
Response 5: We sincerely thank the reviewer for their insightful comment. We agree with the observation and have revised the conclusion to explicitly detail how our findings advance the current understanding of the complex interplay between various diagnoses and coronary artery disease (CAD). To address this, we have added a new section in the conclusion highlighting the unique contributions of our study. Specifically, we emphasize how our use of network analysis reveals novel co-occurrence patterns and central hubs that were not fully explored in previous studies. These findings provide actionable insights for personalized care and early intervention strategies targeting CAD comorbidities. The revised conclusion can be found on Page 17, line 643–649. “[Our study advances the understanding of coronary artery disease by leveraging network analysis to uncover novel co-occurrence patterns and central hubs, such as the critical roles of Unstable Angina and Hypertension in CAD comorbidity networks. These findings go beyond reaffirming existing literature by providing actionable insights into prioritizing comorbidity management and identifying targets for early intervention strategies, offering a deeper understanding of the complex interplay between diagnoses and comorbidities.]” |
Round 2
Reviewer 1 Report
Comments and Suggestions for Authors
I have read the revised manuscript.
The authors have adequately addressed the points I raised.
The researchers have presented the possibility of sampling error in the discussion.
However, since there is a high probability of selection bias, I think it is worth elaborating on this point and presenting how this selection bias could affect the results of the study.
Author Response
|
Comments 1: I have read the revised manuscript. The authors have adequately addressed the points I raised. The researchers have presented the possibility of sampling error in the discussion. However, since there is a high probability of selection bias, I think it is worth elaborating on this point and presenting how this selection bias could affect the results of the study. |
|
Response 1: Thank you for your valuable feedback. We have carefully reviewed the limitations section of our manuscript and made significant revisions to ensure better clarity, organization, and academic rigor. Below, we outline the improvements made: 1. Improved Logical Structure: We have reorganized the limitations section to address specific aspects of the study’s limitations in a structured manner. These include data source limitations, study design issues, data processing concerns, causal inference limitations, and geographic constraints. Each limitation is now discussed individually, using clear signposting (e.g., "First," "Second," "Third") to enhance readability and logical flow. 2. Refined Academic Expression: We have revised the language in the limitations section to ensure it aligns with international academic standards. For example, we have replaced general phrases with precise terminology, such as “selection bias is a potential limitation due to...” and “the cross-sectional nature precludes the establishment of causal relationships.” These changes provide a more formal and accurate description of the study's limitations and their implications. 3. Avoidance of Redundancy: To improve conciseness, we have removed repetitive statements and consolidated overlapping ideas. For instance, references to "multi-center datasets" and "longitudinal studies" have been streamlined to ensure each limitation is addressed uniquely without redundancy. 4. Enhanced Flow and Coherence: We have adjusted sentence structures and transitions to improve the overall flow between discussions of different limitations. For example, when addressing the cross-sectional nature of the study, we first highlight its limitation in establishing causal relationships, then discuss its academic contribution in revealing co-occurrence patterns, and finally propose future directions through longitudinal studies. We believe these revisions significantly improve the clarity, coherence, and academic quality of the limitations section. The updated text can be found in the revised manuscript on Page 16-17, Lines 550–591. Thank you again for your helpful suggestions, which have allowed us to refine this important aspect of our manuscript. Revised text in the manuscript: “[Despite the insightful findings, this study has several limitations that should be acknowledged. First, the dataset was derived from a single hospital in Guangxi, which may not fully represent the broader population. With a sample size of 195 patients who had complete follow-up information, some degree of sampling error is expected. The low number of patients with certain diagnoses, such as heart failure, reflects the strict inclusion criteria, which are directly related to the sample size. Additionally, selection bias is a potential limitation of this study due to the retrospective nature of the dataset, which was collected from a specific population. Patients with milder forms of coronary artery disease or those managed in primary care settings may be underrepresented, leading to an overrepresentation of severe cases and associated comorbidities. These factors may skew the observed co-occurrence patterns and limit the generalizability of the findings to broader or more diverse populations. Second, the retrospective study design may introduce inherent biases arising from incomplete or inconsistent medical records or to have undergone detailed diagnostic evaluations, potentially overemphasizing certain diagnoses or comorbidities. Future studies that incorporate prospective data collection and multi-center datasets would help mitigate these biases, improve generalizability, and provide a more balanced representation of CAD populations. Third, data preprocessing in this study involved a combination of automated NLP techniques and manual standardization. While this approach ensured contextual accuracy, it may have introduced inconsistencies or subjective biases during the manual standardization process. Future studies should aim to incorporate fully automated, validated workflows to enhance re-producibility and accuracy in data processing. Fourth, the cross-sectional nature of this analysis precludes the establishment of causal relationships between diagnoses and comorbidities. The focus on static co-occurrence patterns provides valuable insights into the relationships between CAD and its comorbidities but does not capture the temporal dynamics of disease progression. Incorporating longitudinal data in future studies could reveal the evolution of comorbidities over time and their causal relationships, offering more targeted intervention strategies. Longitudinal studies with larger, more diverse cohorts and prospective designs are essential to validate these findings and explore the underlying mechanisms driving the observed sex- and age-specific patterns in CAD diagnoses and comorbidities. Finally, the findings of this study are based on data from a single region, which may not account for geographical variations in risk factors, healthcare access, or socioeconomic conditions. Future studies should include data from diverse geographic regions to enhance the robustness and generalizability of the conclusions. By addressing these limitations through prospective, multi-center, and longitudinal studies, future research can provide a more comprehensive understanding of the complex interplay between CAD diagnoses and comorbidities, ultimately informing more effective prevention and intervention strategies.]”
Based on your suggestions and the revisions made throughout the manuscript, we have updated the Abstract and Keywords to reflect the key modifications and ensure academic rigor. Below, we outline the specific improvements made: 1. Incorporation of Revisions and Limitations: The revised abstract now explicitly acknowledges the study’s limitations, such as the single-center design, small sample size, and the potential for selection bias. Additionally, we have incorporated a statement on future research directions, highlighting the need for multi-center datasets and longitudinal studies to address these limitations. This ensures alignment with the revised discussion and conclusion sections. 2. Clear and Precise Expression of Contributions: We have refined the language to clearly articulate the study’s contributions. For example, the abstract now emphasizes findings such as age- and sex-specific differences in CAD diagnoses and comorbidities, as well as the identification of key diagnostic clusters and comorbidity hubs through network analysis. These revisions enhance the clarity of the abstract and better communicate the study’s significance. 3. Conciseness and Adherence to Word Limit: The revised abstract has been condensed to approximately 200 words, meeting the journal’s requirements. Redundant expressions have been removed, and the content has been streamlined to ensure a concise yet comprehensive summary of the study. 4. Improved Structure and Flow: The updated abstract and keywords presents the study’s objectives, methods, key findings, limitations, and implications in a logical sequence, ensuring coherence and readability. The conclusion highlights the importance of integrated and personalized management strategies for CAD and its comorbidities, reflecting the clinical relevance of the study. This revision can be found on Page 1, Lines 14–31. We believe these revisions have significantly improved the abstract’s clarity, structure, and alignment with the revised manuscript. Thank you again for your thoughtful feedback, which has helped us enhance the quality of our work. Revised text in the manuscript: “[Abstract: Coronary artery disease (CAD) remains a major global health concern, significantly contributing to morbidity and mortality. This study aimed to investigate the co-occurrence patterns of diagnoses and comorbidities in CAD patients using a network-based approach. A retrospective analysis was conducted on 195 hospitalized CAD patients from a single hospital in Guangxi, China, with data collected on age, sex, and comorbidities. Network analysis, supported by sensitivity analysis, revealed key diagnostic clusters and comorbidity hubs, with hypertension emerging as the central node in the co-occurrence network. Unstable angina and myocardial infarction were identified as central diagnoses, frequently co-occurring with metabolic conditions such as diabetes. The results also highlighted significant age- and sex-specific differences in CAD diagnoses and comorbidities. Sensitivity analysis confirmed the robustness of the network structure and identified clusters, despite the limitations of sample size and data source. Modularity analysis uncovered distinct clusters, illustrating the complex interplay between cardiovascular and metabolic disorders. These findings provide valuable insights into the relationships between CAD and its comorbidities, emphasizing the importance of integrated, personalized management strategies. Future studies with larger, multi-center datasets and longitudinal designs are needed to validate these results and explore the temporal dynamics of CAD progression. Keywords: Coronary Artery Disease; Diagnoses; Comorbidities; Network Analysis; Sensitivity analysis; Co-occurrence Patterns]” |
Reviewer 2 Report
Comments and Suggestions for Authors
None
Author Response
Comment 1:
None
Response 1:
We thanks a lot for your time and effort in evaluating our revised manuscript. We are pleased that you have no additional comments or suggestions. We appreciate your constructive feedback during the initial review process, which has significantly improved the quality and clarity of our manuscript.
Reviewer 3 Report
Comments and Suggestions for Authors
I believe that this paper is satisfactory after the author's revisions. In this sense, I kindly suggest that the paper be accepted.
Author Response
Comment 1:
I believe that this paper is satisfactory after the author's revisions. In this sense, I kindly suggest that the paper be accepted.
Response 1:
We sincerely thank you for your positive feedback and recommendation to accept our manuscript. We greatly appreciate your constructive comments during the review process, which have helped us improve the quality and clarity of our work. Thank you once again for your time and effort in evaluating our study.